# The axonal actin-spectrin lattice acts as a tension buffering shock absorber

**Sushil Dubey[1], Nishita Bhembre[1], Shivani Bodas[2], Sukh Veer[1], Aurnab Ghose[2], Andrew Callan-Jones[3]\*, Pramod Pullarkat[1]\***

[1]Raman Research Institute, Bangalore, India; [2]Indian Institute of Science Education and Research, Pune, India; [3]Laboratory of Complex Materials Systems, Paris Diderot University, Paris, France

**Abstract** Axons span extreme distances and are subject to significant stretch deformations during limb movements or sudden head movements, especially during impacts. Yet, axon biomechanics, and its relation to the ultrastructure that allows axons to withstand mechanical stress, is poorly understood. Using a custom developed force apparatus, we demonstrate that chick dorsal root ganglion axons exhibit a tension buffering or strain-softening response, where its steady state elastic modulus decreases with increasing strain. We then explore the contributions from the various cytoskeletal components of the axon to show that the recently discovered membrane-associated actin-spectrin scaffold plays a prominent mechanical role. Finally, using a theoretical model, we argue that the actin-spectrin skeleton acts as an axonal tension buffer by reversibly unfolding repeat domains of the spectrin tetramers to release excess mechanical stress. Our results revise the current viewpoint that microtubules and their associated proteins are the only significant load-bearing elements in axons.

**\*For correspondence:**
andrew.callan-jones@univ-paris-diderot.fr (AC-J);
pramod@rri.res.in (PP)

**Competing interests:** The authors declare that no competing interests exist.

## Introduction

Axons are micron-thin tubular extensions generated by neuronal cells to transmit signals across large distances — up to the order of centimeters in the brain and up to a meter in the peripheral nervous system of a human body (*Alberts et al., 2002*). In order to achieve such extreme aspect ratios, axons have evolved a unique organisation of the cytoskeleton. It has an axi-symmetric structure with a central core of aligned microtubules arranged in a polar fashion and cross-linked by associated proteins (*Figure 1*). This core is surrounded by neurofilaments and a membrane-associated cortex of cross-linked actin filaments (*Alberts et al., 2002*). Recent experiments using super-resolution light microscopy have revealed that the actin cortex also involves a periodic array of F-actin rings which are interconnected by spectrin cross-bridges (*Xu et al., 2013*; *D'Este et al., 2015*). A myriad of proteins, including motor proteins, interconnect these structural elements to form a dynamic composite gel.

Axons can be subjected to large stretch deformations under a variety of normal as well as abnormal conditions. Mammalian sciatic nerves, for example, can experience localized strains up to 30% during regular limb movements (*Phillips et al., 2004*). A much more extreme case of stretching occurs in the nerves running along the mouth-floor of certain species of baleen whales where it can reach up to 160% during feeding, and these axons have evolved special strategies to cope with such situations (*Lillie et al., 2017*). Recent in vivo studies in humans have shown significant changes in nerve stiffness due to limb flexion (*Andrade et al., 2016*). This suggests that nerve tissue may possess unique mechanical behavior, perhaps to offer some protection against stretch injuries, which are common. It is also known that the brain, being one of the softest tissues, undergoes significant shear deformations (2–5% strain) even under normal activities such as jumping (*Bayly et al., 2005*). At higher accelerations, transient to traumatic brain concussions occur and is a leading cause of

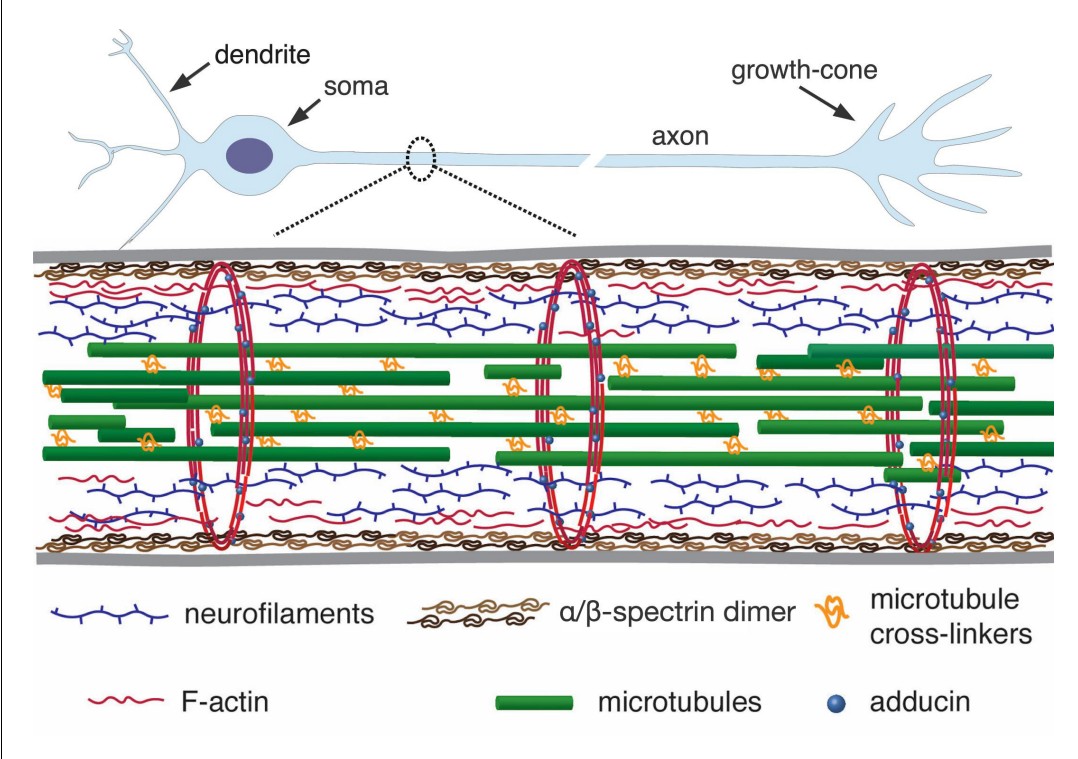

**Figure 1.** A simplified schematic of the axonal cytoskeleton. Typically, the axonal core has a bundle of microtubules which are cross-linked by a variety of microtubule-associated proteins which includes tau (in some cases, a more loose organisation of microtubules interdispersed with neurofilaments is seen) (*Hirokawa, 1982*). This core is surrounded by neurofilaments. The outermost scaffold has an array of periodically spaced rings composed of F-actin filaments. The actin rings are interconnected by $\alpha/\beta$-spectrin tetramers, which are aligned along the axonal axis (only tetramers in a cross-section are shown for clarity). Other cortical F-actin structures also exist. A myriad of proteins, including motor proteins (not shown), interconnect the various filaments, and also the membrane (grey lines) to the inner skeleton. The chick DRG axons we use are about 1 µm thick and the rings in them are about 200 nm apart (corresponds to the length of a single tetramer as shown in Refs [*Xu et al., 2013*; *D'Este et al., 2015*]).

injury in contact sports, with an estimated 1.6 to 3.6 million cases annually in the USA alone (*Harmon et al., 2013*). Hence, knowing how and under what conditions brain deformation compromises axonal integrity is considered essential to understand, manage, and treat concussions (*Johnson et al., 2013*; *Meaney and Smith, 2011*).

At the single-cell level, several studies have explored axonal responses to stretch deformations, primarily using glass micro-needles to pull on live axons and video microscopy to record the evolution of force and strain. Pioneering experiments by Heidemann and co-workers show that axons respond viscoelastically to stretch deformations (*Dennerll et al., 1989*). Besides, they have shown that axons can also develop excess tension when stretched, presumably due to the action of molecular motors (*Dennerll et al., 1989*). Such contractile stress generation has since been studied experimentally and theoretically in some detail and is found to arise due to the action of myosin-II molecular motors (*Bernal et al., 2007*; *O'Toole et al., 2015*; *Tofangchi et al., 2016*; *Mutalik et al., 2018*). Despite the observation of such a rich variety of mechanical responses, the structure–function relationships between the different cytoskeletal elements and these mechanical responses are poorly understood.

In this article, we use a strain-controlled, optical fiber based force apparatus to show that axons exhibit a hitherto unknown strain-softening behavior where the steady state elastic modulus diminishes with increasing strain. By combining our extension rheology technique with biochemical and genetic interventions that alter specific cytoskeletal components, we demonstrate that apart from microtubules, the actin-spectrin skeleton is also a major contributor to the axonal mechanical response to stretch. Then, with the help of a theoretical model, we argue that strain-softening possibly arises from the force-induced unfolding of spectrin subunits, a process known to occur when

single spectrin molecules are stretched (*Rief et al., 1999*). Based on these findings we propose that the actin-spectrin skeleton in axons can act as a tension buffer or 'shock-absorber' allowing axons to undergo significant deformations without the excess build up of tension, as seen in our experiments. Moreover, we show that this mechanism renders axons with a viscoelastic solid-like response, with a memory of the initial state, which allows axons to undergo reversible stretch deformations.

## Results

### Axons exhibit strain-softening, viscoelastic solid-like response

In order to study the axonal response as a function of imposed strain, we applied successively increasing strain steps with a wait time between steps using the home-developed force apparatus shown in *Figure 2A* (image of setup: *Figure 2—figure supplement 1*, stretching of axon: *Figure 2—video 1*; *Seshagiri Rao et al., 2013*). After each step, the strain is held constant using a feedback algorithm. The resulting force relaxation data is shown in *Figure 2B*. Unless specified otherwise, all data are for cells grown for 2 days in vitro (2-DIV). The force relaxation after each step is indicative of the viscoelastic nature of the axon. From the data, we can calculate the axonal tension $\mathcal{T} = F/(2\sin\theta)$ (see *Figure 2C*), where $\sin\theta = d/\sqrt{d^2 + (L_0/2)^2}$, with $\theta(t)$ as the angle with respect to the initial position, $d$ is the displacement of the tip of the cantilever which is in contact with the axon mid point, and $L_0$ is the initial length of the axon. The calculated tension plotted in *Figure 2B* shows that it tends to relax to the same steady state value $\mathcal{T}_{ss}$ as the strain on the axon increases (tension homeostasis). The inset of *Figure 2D* shows this trend for $\mathcal{T}_{ss}$ seen for multiple axons. The occurrence of a steady state tension (or a steady state force) after each strain step is indicative of a solid-like behavior of the axons at longer timescales when probed at room temperature to suppress growth and other active responses. To check this further, we performed a few experiments with wait time ~10 min >> the typical relaxation time (quantified later) after a strain step and these show that the force indeed decays to a non-zero steady state value (*Figure 2—figure supplement 2*).

We note that the axonal structure is highly anisotropic and is composed of multiple cytoskeletal structures, and its elastic response is non-linear. Considering extensile deformations alone, we calculate an 'effective' Young's modulus $E = (\mathcal{T}_{ss} - \mathcal{T}_0)/(A.\gamma)$, where $\mathcal{T}_0$ is the tension of the axon at zero strain which is calculated by extrapolating the tension vs. strain data (*Figure 2—figure supplement 3*), $A$ is the cross-sectional area of the axon (neglecting the change in $A$ with strain; (see Materials and methods for reasoning), and $\gamma = (\sqrt{L_0^2 + 4d^2} - L_0)/L_0$ is the imposed strain. Due to the highly anisotropic nature of the axonal ultrastructure (*Figure 1*), this modulus is expected to be different from that measured by applying local radial deformations using AFM or magnetic tweezers (*Ouyang et al., 2013*; *Grevesse et al., 2015*), where values of 7–10 kPa has been reported. The effective modulus data we obtained from multiple axons are shown in *Figure 2D*, with more examples in *Figure 2—figure supplement 4*. Remarkably, the elastic modulus showsstrain-softening which reflect the tendency for tension to saturate with increasing strain (tension homeostasis). This strain-softening response is not due to any permanent damage or plastic flow as the response remains qualitatively the same when repeated for the same axon. Moreover, when the axons are released from the cantilever from the maximal strain state they recover their initial length within 5–10 s (*Figure 2—video 2*).

### F-actin plays a more significant role than microtubules in axonal response to stretch

Since the turnover of cytoskeletal elements, such as F-actin and microtubules, can be strain-dependent, for example, via the opening of stretch-activated $Ca^{++}$ channels, it is conceivable that the strain-softening response arises due to cytoskeletal remodeling. To test this, and the relative contributions of different structural elements, we performed experiments aimed at either stabilizing or destabilizing microtubules and F-actin using 2-DIV axons. We first tested the response of axons treated with the microtubule stabilizer Taxol at 10 µM for 30 min and the resulting data are shown in *Figure 3A*. The elastic moduli show a strain-softening response similar to normal axons, albeit with higher values of the moduli. Next, in order to check whether F-actin dynamics could lead to strain-softening, we treated neurons with 5 µM of the F-actin stabilizing agent Jasplakinolide for 30 min;

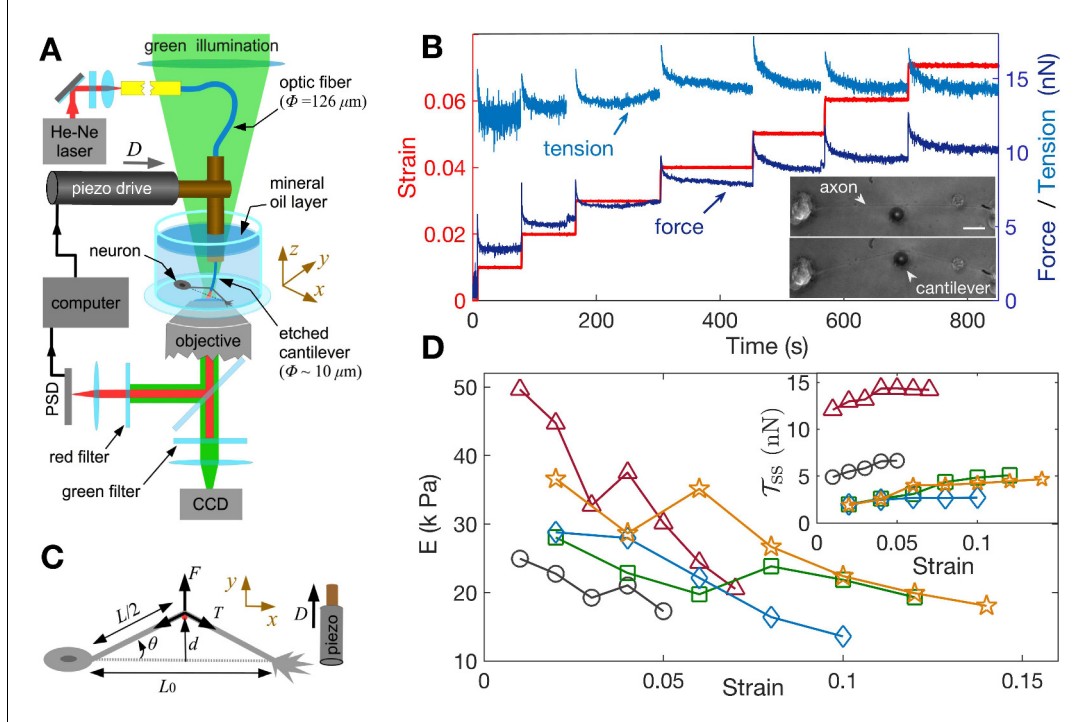

**Figure 2.** Feedback-controlled axon stretch apparatus reveals strain-softening behavior. (**A**) The schematic of the home-developed force apparatus that uses an etched optical fiber as a cantilever to stretch axons and to sense force. Laser light exiting the cantilever tip is imaged on to a Position Sensitive Detector (PSD), which in turn is read by a computer. The computer controls a piezo drive via a feedback algorithm to apply strain steps and then to maintain the strain constant after each step. (**B**) Typical force response (dark blue) of a 2-DIV axon to increasing strain explored using successive strain steps (red). The calculated tension in the axons (sky blue) is also shown. The inset shows the images of the axon before and after stretch (scale bar: 20 µm). The light exiting the etched optical fiber cantilever can be seen as a bright spot (reduced in intensity for clarity), and this is imaged on to the PSD to detect cantilever deflection. (**C**) Illustration of the parameters used in the calculations. The strain is calculated as $\gamma = (\sqrt{L_0^2 + 4d^2} - L_0)/L_0$, and force on the cantilever as $F(t) = -k(D - d)$, where $k$ is the cantilever force constant. The axonal tension is $\mathcal{T} = F/(2\sin\theta)$, where $\sin\theta = d/\sqrt{d^2 + (L_0/2)^2}$. (**D**) Young's moduli calculated for different 2-DIV axons using the steady state tension after each step show a strain-softening behavior (the different symbols are for different axons). Only a few representative plots are shown for clarity, and more examples are shown in *Figure 2—figure supplement 4*. The inset shows the tension vs. strain plots for different axons. Tension tends to saturate with increasing strain (tension homeostasis), which leads to the observed softening.

The online version of this article includes the following video and figure supplement(s) for figure 2:

**Figure supplement 1.** Photograph of the force device.
**Figure supplement 2.** Long time behavior of stretched axons.
**Figure supplement 3.** Determination of rest tension $\mathcal{T}_0$.
**Figure supplement 4.** Normalized moduli for control axons.
**Figure 2—video 1.** Video showing an axon being stretched using the sequential step-strain protocol.
https://elifesciences.org/articles/51772#fig2video1
**Figure 2—video 2.** Video showing an axon being released from the cantilever from a pre-stretched state.
https://elifesciences.org/articles/51772#fig2video2

see *Figure 3B*. As shown in *Figure 3C*, F-actin-stabilized axons exhibit a more substantial increase in the elastic moduli compared to Taxol treatment, while both display pronounced strain-softening. Next, even though our experiments were performed at room temperature to suppress motor activity (*Yengo et al., 2012*; *Hong et al., 2016*), in order to check whether the axonal response is influenced by acto-myosin contractility as reported in *Tofangchi et al. (2016)*; *Mutalik et al. (2018)*, we incubated the cells with 30 µM Blebbistatin for 30 min and checked their response as above (*Figure 3—figure supplement 1*, *Figure 3—figure supplement 2*). No distinction could be seen between treated and untreated cells.

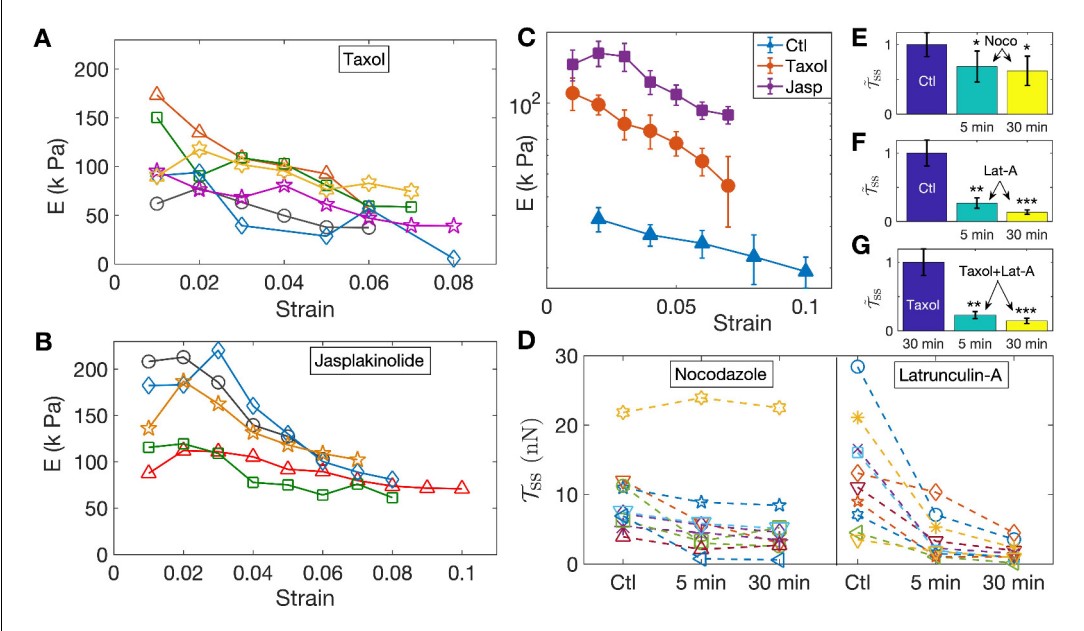

**Figure 3.** Pharmacological perturbations of the cytoskeleton reveal a key role for F-actin in the axonal stretch response. (**A,B**) Stabilizing microtubules or F-actin by treating 2-DIV axons with either 10 µM Taxol or 5 µM Jasplakinolide leads to significant increases in axonal stiffness, but the strain-softening response persists (more data in *Figure 3—figure supplement 6*, *Figure 3—figure supplement 7*; the different symbols are for different axons). (**C**) Log-linear plots of the averaged elastic moduli as a function of strain for different treatments as well as control (for data shown in A,B and *Figure 2D*). It can be seen that stabilization of F-actin causes a larger increase in moduli compared to stabilizing microtubules (error bars are standard error (SE)). (**D**) Change in steady state tension $\tilde{\mathcal{T}}_{ss}$ obtained after treating 2-DIV neurons with either the microtubule disrupting drug Nocodazole (10 µM) or the F-actin disrupting drug Latrunculin-A (1 µM) (n = 10 each). These treatments leave the axons very fragile and they detach easily when pulled. Hence a cyclic step-strain protocol with $\gamma = 0.01$ was employed as detailed in the text. As $\mathcal{T}_0$ cannot be determined from single steps, we compare the net tension for the same axon before (Ctl) and after 5 min and 30 min of treatment. The data show a significant decrease in axonal tension after F-actin disruption and a relatively smaller decrease after disrupting microtubules. (**E, F**) Bar plots of steady state tension normalized by that of control $\tilde{\mathcal{T}}_{ss}$ showing the significance of the two treatments shown in D. Tension was measured for the same axons 5 min and 30 min after exposing to the drug (error bars are SE). (**G**) Bar plots of steady state tension normalized by that for Taxol pre-treatment $\tilde{\mathcal{T}}_{ss}$ for axons initially treated with Taxol for 30 min, and for the same axons subsequently treated with Lat-A for 5 min and 30 min (n = 7).

The online version of this article includes the following figure supplement(s) for figure 3:

**Figure supplement 1.** Force response of Blebbistatin treated cells.
**Figure supplement 2.** Elastic moduli of Blebbistatin-treated cells.
**Figure supplement 3.** Cyclic strain protocol.
**Figure supplement 4.** Force relaxation of Nocodazole-treated cells.
**Figure supplement 5.** Force relaxation of Latrunculin-A-treated cells.
**Figure supplement 6.** Normalized moduli for Taxol treated cells.
**Figure supplement 7.** Normalized moduli for Jasplakinolide treated cells.

To further investigate the possible role of the microtubule and F-actin cytoskeletons, we performed axon stretch experiments after specifically disrupting each of these elements. To avoid detachment of the growth cone or the soma of the drug-treated, cytoskeleton-weakened axons, instead of the previous increasing step-strain experiments, we subjected them to a cyclic strain protocol where repeated up and down steps of equal magnitude are applied (*Figure 3—figure supplement 3*). Moreover, to reduce scatter in data due to natural axon to axon variation, the same axon was probed before and after treatment. We first applied cyclic strain on control axons for extended periods to ensure that axons are not damaged under such conditions (*Figure 3—figure supplement 3*). We then performed measurements after depolymerizing microtubules using Nocodazole (Noco) at 10 µM for up to 30 min. After this treatment, some axons exhibited pronounced beading (Pearling Instability) as is expected when microtubules are lost (*Datar et al., 2019*). As can be seen from *Figure 3D*, on the average, treated axons showed a reduction in steady state tension: $\mathcal{T}_{ss}(\gamma = 0.01) = 9.3nN \pm 1.6(control), \quad 6.4nN \pm 2.1(Noco:5min), \quad 5.8nN \pm 2.0(Noco:30min)$ where the

values are mean ± SE (see *Figure 3E*, and individual data in *Figure 3D*). Next, we disrupted F-actin using 1 µM Latrunculin-A (Lat-A) for up to 30 min and observed that this treatment produced a much more dramatic reduction in the steady state tension than did Nocodazole, as shown in *Figure 3D*. In this case, we obtained, $\mathcal{T}_{ss}(\gamma = 0.01) = 13.0nN \pm 2.5(control)$, $3.5nN \pm 1.0(Lat - A : 5min)$, $1.8nN \pm 0.4(Lat - A : 30min)$, where the values are mean ± SE (see *Figure 3F*, and individual data in *Figure 3D*). This correlates well with the tendency of Jasplakinolide to cause a larger increase in axon elastic modulus as compared with Taxol (see *Figure 3C*). In addition, the tension relaxation is much faster for Latrunculin-A treatment as compared with either Nocodazole-treated or control axons (*Figure 3—figure supplement 4*, *Figure 3—figure supplement 5*), suggesting that F-actin plays a leading role in axonal mechanics.

The larger effects seen after F-actin stabilization or disruption, when compared to microtubule perturbation, comes as a surprise as axonal mechanics is thought to be dominated by microtubules. Since there are interdependencies between the stability of these two components (*Datar et al., 2019*), we tested whether the sharp decrease in elastic modulus after F-actin disruption could be due to microtubules also becoming destabilized subsequent to Lat-A treatment. For this, we first treated axons with Taxol to stabilize microtubules, and then exposed these axons to Lat-A (all concentrations and time as above). This data, presented in *Figure 3G*, shows a drastic reduction in the steady state tension after Lat-A treatment of microtubule stabilized axons. The values are, $\mathcal{T}_{ss}(\gamma = 0.01) = 15.0nN \pm 3.0(Taxol), 3.5nN \pm 0.8(Taxol + Lat - A : 5min)$, $2.2nN \pm 0.6(Taxol + Lat - A : 30min)$. Comparing these results with those without Taxol treatment suggests that actin disruption leading to axon softening is not accompanied by microtubule disassembly.

The emergence of F-actin as more relevant than microtubules to the axon mechanical response under stretching is surprising given that microtubules usually form an aligned and tightly cross-linked bundle at the core of the axons. This finding prompted us to explore in more detail how F-actin regulates the axonal stretch response, in particular via the periodic lattice of rings it forms together with $\alpha$ and $\beta$ spectrin tetramers (*Xu et al., 2013*; *D'Este et al., 2015*).

## Spectrin contributes prominently to axonal stretch response

To check the spectrin distribution along axons we first imaged spectrin content in chick DRG axons using antibody labelling and confocal microscopy. Every axon imaged (n = 180) showed significant spectrin fluorescence distributed all along the axon even at 2-DIV (*Figure 4—figure supplement 1*). The ultra-structure of the spectrin organisation is revealed in the STED nanoscopy images shown in *Figure 4A1*; *Figure 4A2*; *Figure 4—figure supplement 2*. The periodic lattice becomes more prevalent with the number of days the neurons are in culture while maintaining a periodicity in the range of 190 to 200 nm (*Appendix 1—table 1* and *Figure 4—figure supplement 3* for quantification). To check the effect of the different drug treatments on the periodic lattice, we did super-resolution imaging of 2-DIV cells after treating them with the stabilizing drugs Taxol or Jasplakinolide for 30 min. The periodic skeleton is well preserved after treatment with either of these drugs. In vehicle control, 32 out of 36 axons showed rings; after treatment with Jasplakinolide, 28 out of 31 showed rings; and after exposure to Taxol, 31 out of 32 showed rings. After treatment with Nocodazole, 15 out of 23 axons showed rings. We also did treatment with Latrunculin-A and no clear signature of periodic rings could be seen in 17 out of 19 axons. Most notably, the periodicity is lost after Lat-A treatment (see *Figure 4A3*; and also *Figure 4—figure supplement 4* for quantification).

Remarkably, the increase in the prevalence of the actin-spectrin lattice with age quantified in *Appendix 1—table 1* correlates well with the mechanical response, with axons of older neurons showing much higher values of Young's moduli for all strain values as can be seen from *Figure 4B*. The axonal rest tension, too, increases slightly with days in culture from $\mathcal{T}_0(2 - DIV) = 3.7nN \pm 1.2$ (n = 10) to $\mathcal{T}_0(4 - DIV) = 5.3nN \pm 2.2$ (n = 7), where the values are mean ± SE (*Figure 4—figure supplement 5*). The variation of the steady state tension $\mathcal{T}_{ss}$ with strain for 4-DIV cells shows that tension tends to saturate, and is shown in *Figure 4—figure supplement 6*.

Next, we performed knock-down experiments using a specific morpholino (MO) against chick $\beta$-II spectrin. Depletion of $\beta$-II spectrin has been reported to abolish the development of the periodic organization of the actin-spectrin membrane-associated skeleton (*Zhong et al., 2014*). Anti-$\beta$-II spectrin morpholino-treated axons show a dramatic decrease in the steady state tension compared to axons treated with a non-specific morpholino as shown in *Figure 4C*. In some cases, force values

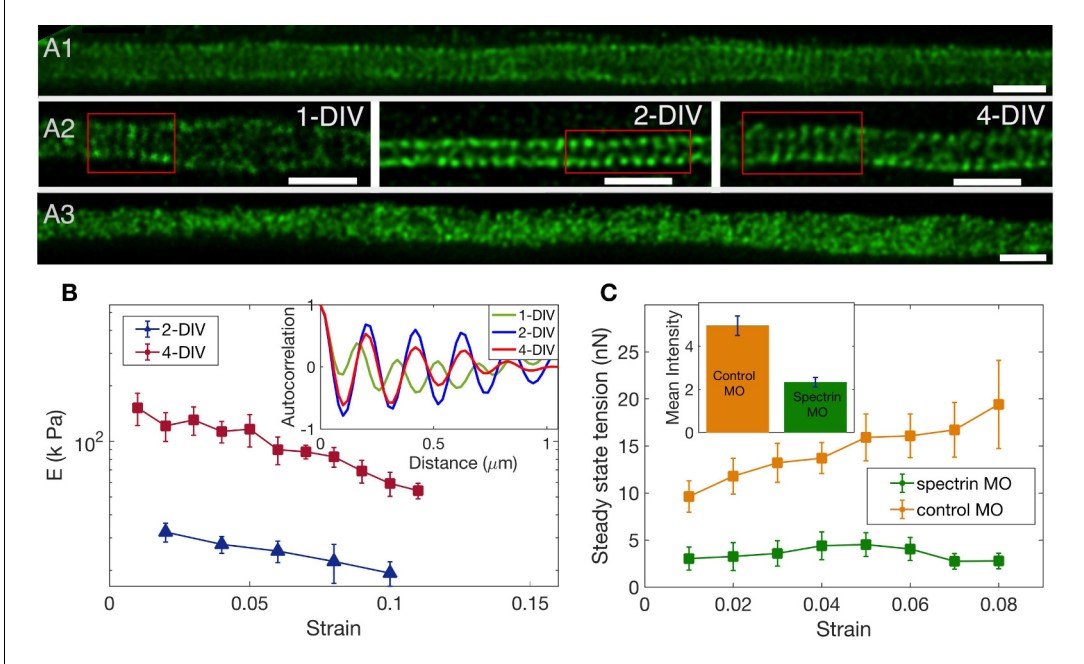

**Figure 4.** The role of spectrin in axonal mechanics. (A1) STED nanoscopy image of a 2-DIV axon immunolabelled with β-II spectrin antibody (vehicle control). This axon shows rings all along the imaged segment (*Figure 4—figure supplement 2* for more examples). (A2) STED images of β-II spectrin in control axons of different DIVs. Observation of a large number of axons shows that the spectrin periodicity becomes more prevalent with the age of the axon (see *Appendix 1—table 1*). (A3) STED image of a 2-DIV axon treated with 1μM Lat-A for 30 min shows disordered β-II spectrin intensity distribution (see *Figure 4—figure supplement 4* for quantification). All scale bars are 1 μm. (B) Log-linear plots of averaged Young's modulus vs strain for axons at 2-DIV (n = 5) and 4-DIV (n = 7) show a marked increase in moduli with age (error bars are SE). This increase in moduli correlated with the prevalence of the actin-spectrin lattice as a function of DIV which is quantified in *Appendix 1—table 1*. The inset shows sample intensity-intensity autocorrelation functions for different DIV axons. The periodicity corresponds to the α/β spectrin tetramer length which is about 190 nm. (C) Plots of the averaged steady state tension $\mathcal{T}_{ss}$ as a function of strain for 2-DIV neurons treated with a non-specific, control morpholino (n = 10) and for those treated with an anti-β-II spectrin morpholino (n = 10). The non-averaged data for control morpholino are shown in *Figure 4—figure supplement 7* for comparison with inset of *Figure 2D*. The averaged data shows that there is a substantial reduction in the tension measured after knock-down. The inset shows the extent of the knock-down obtained using antibody labelling of spectrin for control (n = 90) and knock-down neurons (n = 91); all error bars indicate the standard error (p < 0.05) (see Materials and methods for details).

The online version of this article includes the following figure supplement(s) for figure 4:

**Figure supplement 1.** Spectrin distribution in axons (Confocal image).
**Figure supplement 2.** Spectrin distribution in axons (STED images).
**Figure supplement 3.** Periodicity of spectrin lattice at various DIV.
**Figure supplement 4.** STED super resolution microscopy analysis of Lat-A-treated axons.
**Figure supplement 5.** Rest tension variation with DIV.
**Figure supplement 6.** Steady state tension *vs.* strain in 4-DIV axons.
**Figure supplement 7.** Non-averaged data for steady state tension *vs.* strain in control Morpholino axons.

were so low that determination of rest tension $\mathcal{T}_0$ was not possible. The extent of knock-down was quantified using antibody labeling and the result is shown in the inset of *Figure 4C*, demonstrating a correlation between reduction in spectrin content and strain-dependent steady state tension shown in the main plot. We could not observe any difference in growth characteristics or caliber between knock-down and control cells. Thus, these results clearly demonstrate the significance of the actin-spectrin skeleton in axonal response to stretch deformations. We then turned to theoretical modelling to gain further insight into how this skeleton may contribute to the observed axonal response.

## Folding-unfolding of spectrin buffers axon tension

The axonal cytoskeleton is complex (*Figure 1*; *Leterrier et al., 2017*) and delineating the contributions of the different components is not trivial. It has been generally assumed that bundled microtubules are the main mechanical element in axons (*de Rooij and Kuhl, 2018*; *Ahmadzadeh et al.,*

*2015*). However, our results demonstrate that F-actin and spectrin play a very prominent role in axonal mechanical response. Furthermore, we have shown that axons behave as strain-softening, viscoelastic materials with long time solid-like behavior, unlike the long time flowing state predicted by pure microtubule-based models that invoke cross-link detachments (*de Rooij and Kuhl, 2018*; *Ahmadzadeh et al., 2015*). These observations thus motivated us to consider other dissipative mechanisms, not usually encountered in cytoskeletal remodelling, to develop a coherent theoretical description of axon stretching.

Upon sudden stretching, tension in an axon first increases then relaxes to a non-zero steady-state value (*Figure 2B*), indicating dissipative processes occurring at the microscopic scale. The observation of a steady state tension at long times precludes unbinding processes—such as between spectrin tetramers and actin rings, and unbinding of microtubules-associated or actin-associated crosslinkers—as repeated unbinding events would relax tension to zero. Strain softening and stress relaxation via filament turnover can also be ruled out based on our stabilization experiments summarised in *Figure 3C*. Instead, our experiments suggest that spectrin tetramers play an important role in determining axon tension. Moreover, AFM experiments show that spectrin repeats sequentially unfold under tension (*Rief et al., 1999*). This suggests that dissipation of the energy stored in the folded structure of the protein might contribute to tension relaxation in the axon. However, spectrin may not be the only protein component susceptible to tension-dependent unfolding/refolding. Tau proteins, which crosslink microtubules in axons, unfold under tension (*Wegmann et al., 2011*), resulting in viscoelastic solid-like behavior (*Ahmadzadeh et al., 2014*; *Ahmadzadeh et al., 2015*). Spectraplakins, which interlink actin, microtubules and neurofilaments (*Suozzi et al., 2012*), and neurofilaments themselves (*Aranda-Espinoza et al., 2002*) are other crosslinkers that may unfold under tension. To gain physical insight into the axon response to stretch, we therefore developed a model of tension relaxation arising from unfolding/re-folding of generic protein crosslinkers within the axon cytoskeleton.

We consider a simplified model axon, treated as a cylinder that is homogeneous along its length. As a result, we suppose that each cross-section is pierced by $M$ types of elastic elements in parallel; this is illustrated in *Figure 5A*. Each type of element, representing a type of crosslinking protein with folded and unfolded repeats and labeled by the index $i$, is assumed to have $n_i$ copies in a cross-section. At mechanical equilibrium the total axon tension, $\mathcal{T}$, is uniform, and thus it can be expressed as the sum of tensions acting on the $\sum_{i=1}^{M} n_i$ elements:

$$\mathcal{T} = \sum_{i=1}^{M} n_i T_i. \tag{1}$$

In this equation, $T_i$ is the tension acting on a typical, or average, element of type $i$. The functional form of $T_i$ depends on the choice of the elastic model for the element, which is not a crucial aspect of the problem. Without assuming any particular polymer model for the moment, $T_i$ depends on the ratio of the element's end-to-end extension $\ell_i'$ to its contour, or relaxed, length $\ell_i$. Throughout, primed lengths refer to the end-to-end extension of an element and unprimed lengths refer to its contour distance. We thus expect that $T_i$ decreases with increasing $\ell_i$. This is a first key aspect of the model: $\ell_i$ changes in time because of unfolding and re-folding events, and can thus result in softening. Furthermore, we assume that each elastic element stretches affinely, meaning that the imposed strain $\gamma$ is felt all the way down to the single element level. This implies that $\ell_i' = (1 + \gamma)\ell_{i0}'$, where $\ell_{i0}'$ is the initial length of element $i$ before imposed stretching occurs ($\gamma = 0$); this is illustrated in *Figure 5A*. As a result, we can express each $T_i$ as a function of strain:

$$T_i = T_i(\Lambda); \quad \Lambda \equiv \lambda \ell_{i0}' / \ell_i, \tag{2}$$

where $\lambda = 1 + \gamma$ is the axon deformation.

In the next step, since protein domain unfolding and re-folding will affect its contour length (*Rief et al., 1999*), we propose that the dynamics of the rest length of element $i$ is governed by a rate equation related to unfolding and folding events:

$$\frac{d\ell_i}{dt} = -\nu_{if}(\ell_i - \ell_{if}) + \nu_{iu}(\ell_{iu} - \ell_i). \tag{3}$$

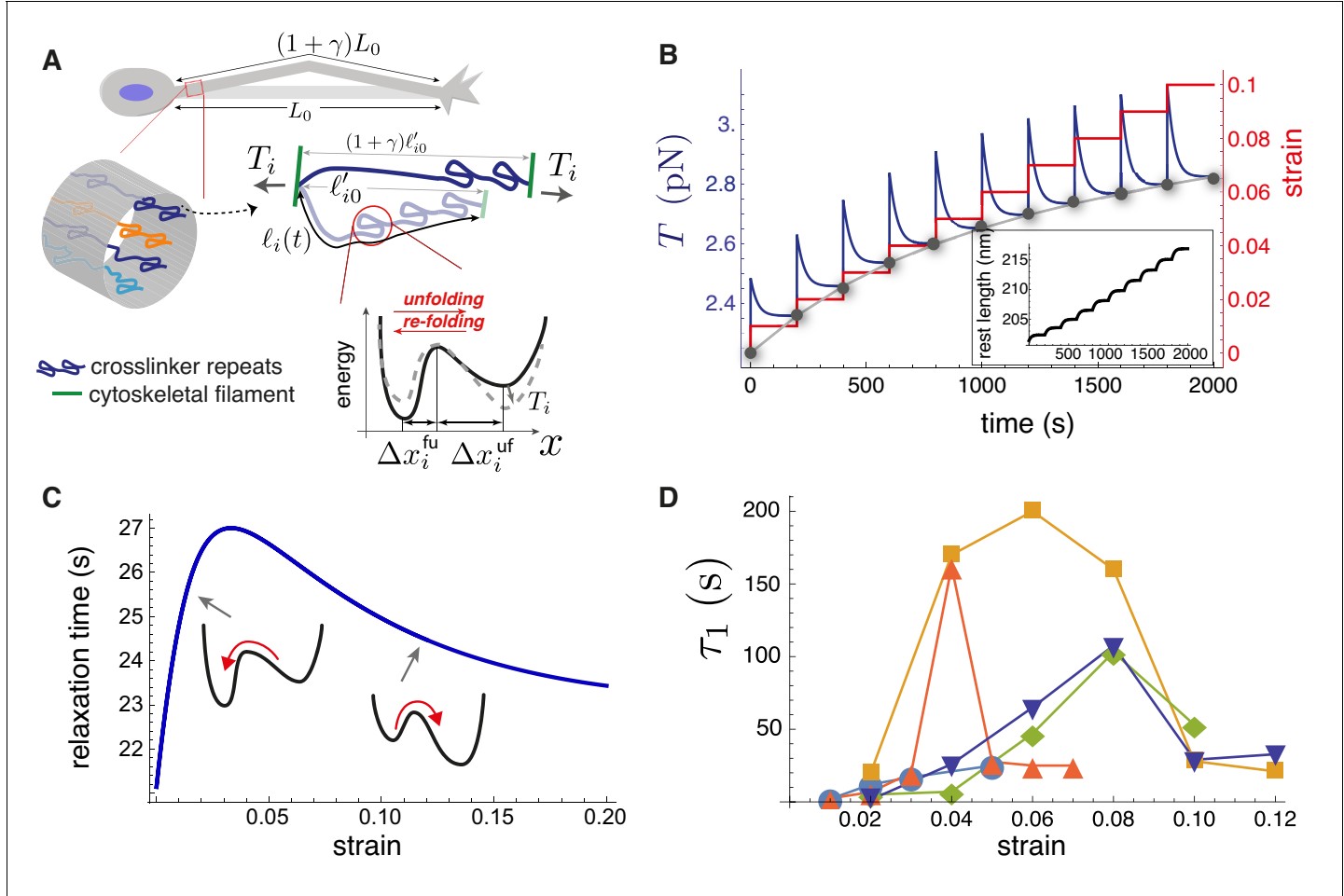

**Figure 5.** Theoretical model recapitulates strain-softening and tension relaxation. (**A**) Schematic illustration of a stretched axon, showing unfolding of a crosslinking protein repeat as an underlying tension relaxation mechanism. The axon, initially with length $L_0$, is stretched to $(1 + \gamma)L_0$. In the model, any cylindrical portion of axon contains various crosslinking proteins (shown using different colors) that experience the strain $\gamma$. The contour length of the crosslinker between two cytoskeletal filaments, $\ell_i(t)$, is variable because of repeat unfolding and re-folding. This process is represented by an energy potential with two minima. The tension, $T_i$, pushes the unfolded minimum down and the folded minimum up. (**B**) Model calculation for a single elastic element for multiple step-strain protocol. Tension versus time (purple) shows a jump after strain increment is applied (red), followed by relaxation to a steady-state value (gray points), passed through by the equilibrium tension versus extension curve "(gray)". This relaxation coincides with progressive repeat unfolding, as represented by the change in rest length $\ell(t)$ (inset). (**C**) Tension relaxation after a small change in applied strain is exponential, with a relaxation time, $\tau$, that depends non-monotonically on the strain. The inset cartoons are to show that $\tau$ is largely controlled by re-folding at low strain, and by unfolding at higher strain. (**D**) Fitting the tension relaxation data obtained from experiments like the one shown in *Figure 2B* to a function that is the sum of two exponentials (*Figure 5—figure supplement 2*, *Figure 5—figure supplement 4*) reveals a long relaxation time (denoted $\tau_1$) with a qualitatively similar dependence on strain as with the model (**C**) and a short relaxation time $\tau_2$ (*Figure 5—figure supplement 3*). Model calculations in B and C were done using the following parameters for spectrin: $\ell_p = 20$ nm (*Stokke et al., 1986*), $\ell'_0 = 170$ nm (*Xu et al., 2013*), $\Delta x^{fu} = 2.5$ nm, $\Delta x^{uf} = 15$ nm, $\ell_u = 1200$ nm, and $\ell_f = 200$ nm. We also used $\nu_f(0) = 100$ s$^{-1}$ and $\nu_u(0) = 10^{-5}$ s$^{-1}$(*Scott et al., 2004*) and $k_BT = 4$ pN.nm.

The online version of this article includes the following figure supplement(s) for figure 5:

**Figure supplement 1.** Model equilibrium tension and effective modulus *vs.*strain.

**Figure supplement 2.** Double exponential fits of tension relaxation data.

**Figure supplement 3.** Shorter relaxation time from double exponential fit.

**Figure supplement 4.** Normalized double exponential fits.

Here, $\nu_{if}$ and $\nu_{iu}$ are rates that represent domain re-folding and unfolding, respectively, on element $i$, and $\ell_{if}$ and $\ell_{iu}$, such that $\ell_{if} < \ell_{iu}$, are the rest lengths in the fully folded and fully unfolded states. We note that $\ell_i - \ell_{if}$ and $\ell_{iu} - \ell_i$ are proportional to the number of unfolded and folded domains, respectively, on element $i$. Generally, we expect that tension accelerates unfolding and hinders re-folding: this is a second key aspect of the model. We assume a molecular scale description

of unfolding and re-folding, in which the folded and unfolded states of a domain are represented by neighboring minima of an energy function; see *Figure 5A*. The effect of the tension on element $i$ is then to tilt the energy landscape down toward the unfolded state, so that in a first approximation (*Zhu and Asaro, 2008*), the rates are given by

$$\nu_{iu} = \nu_{iu}(0)\, e^{\beta T_i \Delta x_i^{\mathrm{fu}}} \tag{4}$$

$$\nu_{if} = \nu_{if}(0)\, e^{-\beta T_i \Delta x_i^{\mathrm{uf}}}, \tag{5}$$

where $\beta = (k_{\mathrm{B}}\mathrm{T})^{-1}$ with $k_{\mathrm{B}}$ the Boltzmann's constant and $\mathrm{T}$ the temperature. In this equation, $\nu_{iu}(0)$ and $\nu_{if}(0)$ are the unfolding and re-folding rates in the absence of tension. The quantities $\Delta x_i^{\mathrm{fu}}$ and $\Delta x_i^{\mathrm{uf}}$ are lengths representing the changes in reaction coordinate in going from the folded to the unfolded state, and vice-versa.

We are now in a position to model the experimental protocol and make sense of our results. For a given choice of function $T_i$ and a given deformation $\lambda$, we can solve *Equations 2, 3, 4 and 5* for the steady-state rest length $\ell_i^{\mathrm{ss}}$, yielding the relation $\ell_i^{\mathrm{ss}} = (\nu_{iu}\,\ell_{iu} + \nu_{if}\,\ell_{if})/(\nu_{iu} + \nu_{if})$. Note that this equation must be solved numerically since the rates depend on $\ell_i^{\mathrm{ss}}$ through *Equations 2 and 4, 5*. Next, at a chosen time denoted $t = t_{\mathrm{step}}$, we apply a small but sudden change in strain, $\delta\gamma$. Right after the step, the tension jumps to a value $T_i(t = t_{\mathrm{step}}) = T_i((\lambda + \delta\gamma)\ell_{i0}'/\ell_i^{\mathrm{ss}})$. The tension will then decay as $\ell_i$ increases from $\ell_i^{\mathrm{ss}}$ due to unfolding. By solving for the linearized relaxation behavior, we can show that the total tension relaxes as a sum of exponentials (see Appendix-1):

$$\mathcal{T}(t) = \sum_{i=1}^{M} n_i T_i(t = t_{\mathrm{step}})\, e^{-(t - t_{\mathrm{step}})/\tau_i}, \tag{6}$$

with relaxation times $\tau_i$ that depend on the strain prior to the small jump. To proceed further, we consider only one type of element and henceforth drop the label $i$. We can then show that the relaxation time is given by

$$\tau = \left[ (\nu_{\mathrm{u}} + \nu_{\mathrm{f}}) + \frac{\nu_{\mathrm{u}}\nu_{\mathrm{f}}}{(\nu_{\mathrm{u}} + \nu_{\mathrm{f}})} \frac{(\ell_{\mathrm{u}} - \ell_{\mathrm{f}})}{\ell} \beta(\Delta x^{\mathrm{fu}} + \Delta x^{\mathrm{uf}})\Lambda K(\Lambda) \right]^{-1}, \tag{7}$$

where $K(\Lambda) = dT(\Lambda)/d\Lambda$ is the differential stiffness; see Appendix-1 for details. In *Equation 7* all strain-dependent quantities are evaluated at the steady-state condition holding prior to $t = t_{\mathrm{step}}$. Also, we note that the first term in this equation is what one would obtain if the unfolding and re-folding rates were constant. The second term accounts for the dependence of these rates on the relaxed length of the element. Next, we use the worm-like chain (WLC) approximation for the element tension $T$ (*Marko and Siggia, 1995*), which has been used previously to model spectrin unfolding (*Rief et al., 1999*):

$$T = \frac{1}{\beta\ell_p} \left( \frac{1}{4(1 - \lambda\ell_0'/\ell)^2} + \frac{\lambda\ell_0'}{\ell} - \frac{1}{4} \right), \tag{8}$$

where $\ell_p$ is the persistence length, and is assumed to be independent of the folding state. This model predicts strain *stiffening* for constant rest length, that is, the slope $\partial T/\partial\lambda$ increases with $\lambda$. It is thus a useful test case to see to what extent the relaxation of the rest length $\ell(t)$ can lead to softening.

Our model can qualitatively reproduce the experimental axon stretch response (*Figure 2B*). During a jump in strain, the tension rises quickly and then relaxes, as crosslinking protein repeats unfold; see *Figure 5B*. We use values of $\ell_p$, $\ell_0'$, $\ell_{\mathrm{f}}$, $\ell_{\mathrm{u}}$, $\Delta x^{\mathrm{fu}}$, $\Delta x^{\mathrm{uf}}$, $\nu_{\mathrm{f}}(0)$, and $\nu_{\mathrm{u}}(0)$ appropriate for spectrin; see Appendix-1 for justification. At long times after each strain step, the tension tends to a steady-state value such that the locus of values follows the steady-state tension versus strain curve, $T_{ss}$ found by solving *Equation 3, 4, 5 and 8* (gray dots in *Figure 5B*). We note that the model predicts that the crosslinker is under tension for zero applied strain, as is the case for the entire axon (*Figure 4—figure supplement 5*). Importantly, the $T_{\mathrm{ss}}$ curve reflects the tension buffering behavior seen experimentally (*Figure 2B* (Tension), 2D(inset), *Figure 4—figure supplement 6*, and Ref [*Rief et al.,*

*1999*]). This buffering, or softening, arises in the model because of a significant increase in the rest length $\ell(t)$ for strains not exceeding 10% (inset of *Figure 5B*, *Figure 5—figure supplement 1*). Taken together, our hypothesis of tension-assisted unfolding of crosslinking proteins can account for the axon strain softening seen experimentally.

With our model, we also predict a surprising, non-monotonic dependence of the tension relaxation time $\tau$ on strain; see *Figure 5C*. This behavior can be understood as follows. For small strain, we expect that the energy minimum of the folded state is much lower than the minimum of the unfolded state. As a result, we expect then that $\nu_u \ll \nu_f$; this disparity has been measured experimentally (*Scott et al., 2004*). Since $\nu_f$ decreases with tension, according to *Equation 7*, the relaxation time is then $\tau \approx \nu_f^{-1}$, which increases with strain, for small strain. With increasing strain the last term in *Equation 7*, which is $\propto \nu_u^{-1} \propto e^{-\beta T \Delta x}$, becomes important, and leads to a decreasing relaxation time with strain. Remarkably, fitting our axon tension versus time data for a multi-step strain protocol (*Figure 2B* and *Figure 5—figure supplement 2*) with a function $A\,e^{-(t-t_{\text{step}})/\tau_1} + B\,e^{-(t-t_{\text{step}})/\tau_2} + C$ we see that both relaxation times $\tau_1$ and $\tau_2$ first increase then decrease with strain; see *Figure 5D*, and *Figure 5—figure supplement 3*. This result is in full qualitative agreement with our model, which is predicated on multiple types of crosslinking proteins in parallel, thus multiple tension relaxation times (*Equation 6*), each with a similar non-monotonic dependence on the strain.

## Discussion

To summarize our main results, using a custom-built, controlled-strain force apparatus, we demonstrate for the first time, to our knowledge, that (i) axons exhibit a strain-softening behavior due to their ability to buffer mechanical tension when stretched (tension homeostasis); and (ii) that their passive response behave as viscoelastic solid, with a relaxation time that depends non-monotonically on the strain. We have, furthermore, unravelled hitherto unknown connections between the axon mechanical response and its cytoskeleton, by using experiments that either stabilize or de-stabilize specific elements, including the actin-spectrin skeleton. We show that apart from microtubules, (iii) F-actin and spectrin emerge as prominent contributors to the axon mechanics. As axons mature, the actin-spectrin periodic skeleton becomes more prevalent and this correlates with an increase in the Young's moduli of the axons with age.

An order of magnitude estimate of the maximum possible contribution coming from spectrin tetramers to axonal tension can be made as follows. If we assume the width of a spectrin molecule to be 10 nm, and take a typical axon diameter as 1 μm, the maximum number of spectrin molecules (for a mature axon, say) in a cylindrical cross-section can be estimated to be about 300. A more conservative estimate can be made using available data from RBCs (*Liu et al., 1987*), where the actin-spectrin junction is reported to be 35 nm in width (with each junction connecting to an alpha and a beta spectrin molecule), and thus a number closer to 180. AFM pulling experiments on native spectrin showed that the force-extension curve first rapidly increased (due to entropic stiffening), and then, due to repeat unfolding, underwent a succession of sharp peaks and drops, bound below by a gently increasing, plateau-like force of about 30 pN (*Rief et al., 1999*). From this, we can estimate the spectrin contribution to axonal tension to be of the order of $\mathcal{T}_{ss} \sim 6-10$ nN. These values are comparable to the axon tension for 4-DIV axons (*Figure 4—figure supplement 6*). Although this is only a rough estimate, and the exact number may depend on the details of the actin-spectrin skeleton, it suggests that spectrins could make a significant contribution to axonal mechanics.

Our experiments show that F-actin contributes significantly to axonal Young's modulus. In our model, the actin rings act as structures that link spectrin tetramers along the axon (*Figure 1*). The rings, being in a plane orthogonal to the stretch direction, are assumed to not undergo any deformation and only the spectrin molecules increase in length. Thus, the rings are vital for mechanical continuity of the actin-spectrin skeleton but are not modeled as an additional elastic element. This is consistent with the above mentioned experimental observation.

A part of the mechanical stress is expected to be borne by microtubules, which form bundles cross-linked by Microtubule Associated Proteins, tau being a major component of this in axons. Assuming affine deformation of the microtubule bundle, we can estimate the relative sliding between adjacent microtubules as $\delta x = (L - L_0)/n$, where $n$ is the number of microtubules needed to span the length of the axon. Taking an average microtubule length of 4 μm (*Yu and Baas, 1994*), a

typical axon length of $L_0 = 200$ µm, and an applied strain of 10%, we get a relative displacement for microtubules as $\delta x = 400$ nm. Taking tau to be the major cross-linking protein in axons, and a relaxed tau-tau dimer length of $l_0 = 80$ nm, each dimer is stretched by an amount $\sqrt{l_0^2 + (\delta x/2)^2} - l_0 \simeq 135$ nm. At such displacements, tau proteins may either stretch without dissociating or stretch and dissociate depending on the loading rate and strain, as modeled in detail in Ref (*Ahmadzadeh et al., 2015*). It has been argued that such a bundle cannot sustain load over long times as it behaves as a fluid by allowing microtubules to slide (*Ahmadzadeh et al., 2015*). This would mean that microtubules do not contribute to the steady state tension. There is also experimental evidence that microtubules are disrupted under fast loading reaching high strains (*Tang-Schomer et al., 2010*). However, in our experiments, we observe an increase in steady state modulus when microtubules are stabilized and a decrease when they are disrupted. This may mean that other more permanent MAPs need to be included. If these are indeed present, stiffening behavior can occur (*Peter and Mofrad, 2012*). Another possibility is that, as shown in Refs (*Zhong et al., 2014*; *Qu et al., 2017*), disruption of microtubules using Nocodazole causes a decay of the actin-spectrin skeleton and stabilizing the filaments using Taxol increases the occurrence of this periodic scaffold.

Neurofilaments (NFs) form another important component to consider in axonal bulk mechanics. These filaments interact electrostatically and very little is known about associated crosslinkers. In vitro experiments show that they can occur as gel-expanded or gel-condensed states (*Beck et al., 2010*), and exhibit strain stiffening at very high crossover strains of 30–70% (*Storm et al., 2005*; *Yao et al., 2010*). However, AFM experiments on neurofilaments show that their sidearms can unfold (*Aranda-Espinoza et al., 2002*), suggesting a role for NFs in softening. Nonetheless, it remains unknown to what extent, if any, NFs contribute to bearing axonal tension. Experiments on axons using magnetic tweezers suggest that these filaments are more viscous than elastic (*Grevesse et al., 2015*). In our model, we have left open the possibility that NFs can contribute to softening via unfolding and re-folding events.

Our experiments and theoretical analysis suggest that protein crosslinkers that can unfold and re-fold, such as spectrin tetramers, can contribute to strain-softening via their force-dependent unfolding and re-folding kinetics. This model predicts key experimental features like the viscoelastic solid-like response and the non-trivial dependence of the tension relaxation time, which first increases and then decreases at larger strains. These two responses are signatures of unfolding and re-folding in tension relaxation, and argue against a few other possible mechanisms mentioned in *Appendix 1—table 2*. For example, a tension relaxation scenario involving, say, actin-spectrin unbinding would lead to a relaxation time that monotonically decreases with strain and a long time fluid-like response. A recent computational model of the actin-spectrin skeleton in axons showed that a spectrin tetramer, once unbound from a ring, cannot rebind (*Zhang et al., 2017*). An additional feature of the force-assisted unfolding of spectrin tetramers with multiple repeat units is that the tension versus strain response at steady-state exhibit an extended region where the tension is only weakly dependent on the strain (*Figure 2D*-inset, *Figure 4—figure supplement 6* and *Figure 5—figure supplement 1*). This can have important functional significance as the actin-spectrin skeleton can protect axons against stretch deformations by acting as a tension buffer, or 'shock absorber'.

There are in-vivo studies that reveal the importance of spectrin in axonal mechanical stability or tension. Notably, in *C. elegans*, the axons of spectrin knock-out animals are known to snap during normal wiggling of the worm, although the exact nature of axonal deformation (buckling vs stretch) that causes damage is unclear (*Hammarlund et al., 2007*). Moreover, the use of a FRET-based spectrin tension sensor has shown that $\beta$-spectrin in *C. elegans* axons are held under pre-stress; and this is further supported by axonal retraction observed after laser ablation (*Krieg et al., 2014*). Still, to date, there have been no studies that quantitatively relate the actin-spectrin axon architecture to its strain response. In contrast, spectrin-mediated elastic behavior leading to strain softening has been documented in experimental and theoretical studies on Red Blood Cells (RBCs), which have a membrane-associated hexagonal actin-spectrin skeleton (*Lee and Discher, 2001*; *Gov, 2007*; *Zhu and Asaro, 2008*; *Li et al., 2007*). The current view is that RBC softening is most likely due to the unfolding of spectrin domains under force (*Johnson et al., 2007*; *Krieger et al., 2011*).

## Strain stiffening and softening in other cell types

The axonal response reported here is in sharp contrast with the strain stiffening response exhibited by many cell types (*Fernández et al., 2006*; *Kollmannsberger and Fabry, 2011*). This stiffening has its origin in the entropic nature of the F-actin cytoskeleton and has been observed in purified systems of a variety of biopolymers (*Storm et al., 2005*). It is also known that the stiffening response can transition toward a softening response at higher strains, either due to the force-induced detachment of transient cross-links or due to buckling of filaments (*Gardel et al., 2004*; *Chaudhuri et al., 2007*; *Åström et al., 2008*). In contrast, the axons we have studied here show only a softening response even at the smallest of explored strain values. It has been reported that eukaryotic cells show a transient softening following a fast stretch and release protocol (*Trepat et al., 2007*). This is a transient effect and is not reflected in the steady state modulus of the cell and has been attributed to ATP driven processes, which make the cell behave as a soft-glassy system (*Trepat et al., 2008*). The same cells, when subjected to a step strain protocol to measure the steady state modulus, exhibit strain stiffening (*Trepat et al., 2007*).

In conclusion, we have demonstrated a mechanical role for the actin-spectrin skeleton in axonal response to stretch. By combining quantitative experiments and theoretical modelling, we show that axonal strain-softening could arise from this spectrin skeleton and it allows axons to undergo significant reversible deformations by acting as molecular bellows which buffer tension. Such processes can increase the resilience of axons to sudden stretch deformations that occur in vivo under normal conditions, as mentioned in the introduction. Although our modeling is restricted to cytoskeletal elements that undergo unfolding-refolding dynamics, how the composite structure of the axon, with its different interdependent cytoskeletal elements, responds to stretch as a whole is an interesting area for further research and modelling. Moreover, how the actin-spectrin skeleton self-assembles and how it may dynamically reorganize under stretch on long timescales during growth will be interesting to explore. Apart from revealing the unique bio-mechanical properties, these results should also motivate the design of novel biomimetic materials which can deform at constant stress while retaining the memory of its initial state. From a medical point of view, spectrin mutations are associated with neurological disorders like spinocerebellar ataxia (*Ikeda et al., 2006*) and early infantile epileptic encephalopathies (*Wang et al., 2018*). Hence, apart from direct mechanical effects, how axonal integrity may be affected by spectrin specific mutations will be an area of interest for future studies.

## Materials and methods

### Key resources table

| Reagent type (species) or resource | Designation | Source or reference | Identifiers | Additional information |
|---|---|---|---|---|
| Genetic reagent | control MO | Standard control oligo, GeneTools | Gene Tools, LLC | 5′-CCTCTTACCTCAGTTACAATTTATA-3′ |
| Genetic reagent | Photo control MO | Standard control oligo, GeneTools | Gene Tools, LLC | 5′-CCTCTTACCTCAGTTACAATTTATA-3′ |
| Genetic reagent | Specific MO | Specific beta-II spectrin Gene tools | Gene Tools, LLC | 5′-GTCGCCACTGTTGTCGTCATC-3′ |
| Antibody | Beta-II spectrin antibody | BD biosciences | RRID:AB_399854 | (1:1500) |
| Antibody | anti-mouse Alexa Fluor 568 | Thermo Fisher Scientific | RRID:AB_2534072 | (1:1000) |
| Antibody | Anti-mouse Alexa Fluor 488 | Thermo Fisher Scientific | RRID:AB_2534069 | (1:1000) |
| Chemical compound, drug | Nocodazole | Sigma-Aldrich | M1404, Sigma-Aldrich | 10 µM |
| Chemical compound, drug | Latrunculin-A | Sigma-Aldrich | L5163, Sigma-Aldrich | 1 µM |
| Chemical compound, drug | Jasplakinolide | Thermo Fisher Scientific | J7473, Thermo Fisher Scientific | 5 µM |

*Continued on next page*

*Continued*

| Reagent type (species) or resource | Designation | Source or reference | Identifiers | Additional information |
|---|---|---|---|---|
| Chemical compound, drug | Paclitaxel (Taxol) | Sigma-Aldrich | T7402, Sigma-Aldrich | 10 µM |
| Chemical compound, drug | Blebbistatin | Sigma-aldrich | B0560, Sigma-Aldrich | 30 µM |

## Primary neuronal culture

Fertilised Giriraja-2 chicken eggs were obtained from Karnataka Veterinary, Animal and Fisheries Sciences University, Bangalore, India. The eggs were incubated for 8–9 days and Dorsal Root Ganglia (DRG) were isolated and dissociated using a standard protocol. Cells were then plated on well-cleaned glass coverslips with an attached cylindrical glass ring (1 cm high, 12 mm dia.) to contain the culture medium. We used L-15 medium (21083–027, Thermo Fisher Scientific) thickened with 0.006 g/ml methyl cellulose (34516, ColorconID) and supplemented with 10% Fetal Bovine Serum (10100–147, Thermo Fisher Scientific), 2% D-glucose (G6152, Sigma-Aldrich), 20 ng/ml Nerve Growth Factor NGF-7S (13290–010, Thermo Fisher Scientific), and 0.5 mg/ml Penicillin-Streptomycin-Glutamine (100×) (10378–016, Thermo Fisher Scientific). Cells were incubated at 37 ˚C for 48 or 96 hr as per experiment. Prior to force measurement experiments, neurons were incubated for 30 min in supplemented L-15 medium lacking methyl cellulose.

## Cytoskeleton treatments

All cytoskeleton perturbing agents such as Nocodazole (M1404, Sigma-Aldrich), Latrunculin-A (L5163, Sigma-Aldrich), Jasplakinolide (J7473, Thermo Fisher Scientific), Blebbistatin (B0560, Sigma-Aldrich), and Paclitaxel (Taxol) (T7402, Sigma-Aldrich) were dissolved in DMSO. The final DMSO concentration was kept well below 1% during experiments. The concentrations of these agents are indicated in the text describing these experiments.

## Functional knockdown of $\beta$-II spectrin

Primary chick DRG neurons were cultured in 400 µl of supplemented L-15 medium as described above for 2–4 hr at 37 ˚C. Neurons were transfected by adding a pre-mixed solution of 20 µM Morpholino oligomers (MO) and 2 µM Endoporter (EP) (GeneTools, LLC) to the culture medium and incubated at 37°C for 48 hr in well humidified chambers. The transfection conditions were optimized by fluorescence microscopy using a carboxyorescein tagged control MO (5'-CCTCTTACCTCAGTTA-CAATTTATA-3'). All neurons (n = 46) showed diffuse fluorescence in the soma and all along the axon indicating the presence of tagged MO in the cytosol. An unlabeled translation blocking MO targeted against $\beta$-II spectrin (5'-GTCGCCACTGTTGTCGTCATC-3') was used for functional knockdown studies. An unlabeled non-specific morpholino (5'-CCTCTTACCTCAGTTACAATTTATA-3') was used as a control. To quantify the extent of $\beta$-II spectrin knockdown, neurons transfected with either anti-$\beta$-II spectrin or control MOs for 48 hr were rinsed in HBSS (with $Ca^{++}$ & $Mg^{++}$), fixed and immunostained as described below. For force measurements on MO transfected cells, the culture medium was replaced by supplemented L-15 lacking methyl cellulose 30 min prior to the experiments.

## Immunofluorescence

Neurons were fixed using 4% (w/v) paraformaldehyde (PFA) and 0.5% (v/v) glutaraldehyde in phosphate buffered saline (PBS: 5.33 mM $KCl$, 0.44 mM $KH_2PO_4$, 4.16 mM $NaHCO_3$, 137.93 mM $NaCl$, 0.33 mM $Na_2HPO_4$, 5.55 mM D-Glucose) for 10 min at room temperature (RT). Neurons were permeabilized with 0.2% (v/v) Triton X-100 in PBS for 5 min at RT and incubated in a blocking buffer of 3% (w/v) Bovine Serum Albumin in PBS at RT for 1 hr. The fixed neurons were rinsed three times with PBS and incubated with anti-$\beta$-II spectrin antibody (BD Biosciences, 612563; 1:1500 dilution in blocking buffer) overnight at 4 ˚C. After rinsing again, the neurons were incubated with anti-mouse IgG conjugated to fluorescent Alexa Fluor 568 (A-11004, Thermo Fisher Scientific) or Alexa Fluor 488 (A-11001, Thermo Fisher Scientific) used at 1:1000 dilution in the blocking buffer for 1 hr at RT. Prior to mounting for microscopic examination, the cells were rinsed again and post-fixed with 4% PFA and 0.5% glutaraldehyde in PBS for 10 min at RT. For super-resolution microscopy, the neurons

cultured for various days in vitro (DIV) were mounted in Mowiol (10% Mowiol 4–88 in poly(vinyl alcohol), 81381, Sigma-Aldrich) and DABCO (2.5% w/v, 1,4-diazobicyclo[2.2.2]octane, D27802, Sigma-Aldrich) and kept overnight in the dark at 4 ˚C prior to imaging.

## Microscopy

$\beta$-II spectrin immunofluorescence of morpholino transfected neurons were evaluated using a confocal laser scanning microscope (Leica TCS SP8, Leica Microsystems) with a 63×, 1.4NA oil immersion objective. All imaging parameters were held constant for control and anti-$\beta$-II spectrin MO transfected neurons. Stimulated Emission Depletion (STED) nanoscopy was carried out using a Leica TCS SP8 STED system (Leica Microsystems). The fixed and mounted samples were imaged with the STED WHITE oil objective lens (HC PL APO 100×/1.40 OIL). CMLE JM deconvolution algorithm from Huygens Professional software (version 17.04) was used for deconvolution and processing of images.

## Image analysis and statistics

The average fluorescence intensity was measured using the segmented line tool of ImageJ software (NIH, USA) with a specified width to trace the axon of interest and to calculate the intensity per unit area of the axon. When doing this, the background intensity per unit area was measured and subtracted from the axon intensity. Origin software (OriginLab, version 9) was used for statistical analysis and graphical representations. The periodicity of $\beta$-II spectrin distribution was evaluated by plotting the intensity trace along the axon using Fiji/Image J. The intensity values were analyzed using the autocorrelation (autocorr) function of Matlab (MathWorks, R2018b). For 1-D autocorrelation analysis, several 1 μm segments were taken from each axon. A line of 1 μm was drawn with a line width on the edge of the axon and the intensity profile is obtained using Image J. For these segments, 1-D autocorrelation was calculated using Matlab. Then an average of the autocorrelation was obtained for all axons. The amplitude of autocorrelation is defined as the difference between the maxima (~200 nm(first peak)) and minima (~100 nm) of the 1-D autocorrelation. The average amplitude is obtained by averaging over all the axons.

All data are represented as mean ± standard deviation of mean (SE) from independent experiments, unless otherwise specified. Data shown represent the number of neurons (n) analysed. Whenever data sets are small (due to practical difficulties in obtaining sufficient sample size), individual data is shown before presenting mean and SE. To avoid errors due to axon to axon variations, trends (variation with strain or with drug treatments) are shown for the same axon whenever possible. For the quantification of spectrin knock-down, statistical significance of difference in mean fluorescence intensity across pooled data sets was tested using non parametric two-tailed Mann Whitney U (MW - U) test with $p < 0.05$ set as the minimum level of significance. Origin software (OriginLab,version 9) and Matlab 2018b were used for all the statistical analyses and graphical representations.

## Force measurement

Measurements were made using a modified version of an optical fiber based force apparatus we had developed earlier (*Seshagiri Rao et al., 2013*), and is shown schematically in the main text *Figure 2A*. It consists of a cylindrical glass cantilever, 10–20 μm in diameter and 5–10 mm in length, fabricated by uniformly etching the end portion of a 126-μm-thick single mode optical fiber. The exact length and diameter of each cantilever is measured and its force constant determined by treating it as a perfect cantilever (*Seshagiri Rao et al., 2013*). We use the Young's modulus value for the optical fiber obtained by loading test cantilevers by known weights and measuring the tip deflection using a horizontal microscope. The base of the cantilever was attached to a closed-loop linear piezoelectric drive (P-841.60, Physik Instruments) which has an accuracy of 1 nm and a travel range of 90 μm. The piezo was mounted on a Zeiss AxioObserver D1 (Carl Zeiss GmbH) microscope using a joystick operated XYZ-stage (XenoWorks, Sutter Instruments). The position of the tip of the cantilever is measured with a resolution of 35 nm by focussing the laser light exiting the fiber on to a Position Sensitive Detector (PSD) (S2044, Hamamatsu). The green microscope illumination light and the red laser light were separated using appropriate filters to enable simultaneous force measurements and imaging using a CCD camera (Andor Luca R604, Andor Technology) and a 40×, 0.5NA LD-A-Plan objective.

The setup was tested using another cantilever as mock sample and this gave the expected linear elastic response. Once the cantilever is placed in the container with cells, a drop of mineral oil is added on top of the culture medium to minimize convection currents due to evaporative cooling. Axons were then pulled laterally at their mid-points by extending the piezo by a distance $D$ as shown in *Figure 2A* of the main article. During this process we ensure that the cantilever and the axon (except at the soma and the growth-cone) are maintained slightly away from the surface of the coverslip to avoid the cantilever touching the surface during experiments. This is then verified by checking the focal position of the cantilever tip and the coverslip surface (identified by tiny debris always present on the surface). Observation of free thermal fluctuations of the cantilever and axon too ensures this. The cantilever deflection during stretching experiments was then calculated as $D - d$ (see *Figure 2C* of the main article). The axonal strain $\gamma$ is calculated from the initial length of the axon $L_0$. A feedback algorithm implemented using LabView (National Instruments, v14.0) calculates the strain steps such that there is no overshoot, and maintains the strain constant for a prescribed wait time after each step. Axons with initial length in the range of about 100–200 μm were chosen for experiments. The axonal diameter was measured in each case using phase-contrast microscopy. We ignore the small change in radius due to stretching as this is below the resolution of the microscope, and a simple estimate of it by assuming a constant axonal volume gives a fractional change in radius $\frac{\Delta r}{r_0} = -0.5 \frac{\Delta L}{L_0} \approx 0.07$. This small decrease in radius would correspondingly cause a small *increase* in Young's modulus, $E$, and cannot account for the softening we observe.

There are practical difficulties in performing stretching experiments on primary neurons. The cell body is poorly adherent and can easily move or detach during mechanical perturbations, especially after treating with cytoskeleton modifying drugs. This limits both the range of strain values and the duration of measurements at each strain. For these reasons, we used successive strain steps with a strain-dependent wait time when probing the non-linear response of the axon and small amplitude cyclic strains for cytoskeleton perturbation experiments. Moreover, active contractile responses to mechanical perturbations and growth cone dynamics can get superposed with the passive force relaxation process. In order to suppress active dynamics as much as possible, we performed all experiments at room temperature (25–26 °C) (*Yengo et al., 2012*; *Hong et al., 2016*). Axons were chosen such that their entire length was within the field of view and the data was discarded if either the soma or the growth cone moved during measurement.

## Acknowledgements

We acknowledge Seshagiri Rao for his help in improving the setup, Jagruti Pattadkal for performing the preliminary experiments, and Serene Rose David for her help with the preparation and use of reagents. We thank the IISER-Pune imaging facility for STED nanoscopy; Deepak Nair and Siddharth Nanguneri for discussions and preliminary trials on super resolution imaging. We thank Andrea Parmeggiani for valuable discussions; N V Madhusudana, Igor Muševič, Girish Deshpande, and Patricia Bassereau for their critical reading of the manuscript. AG and PP acknowledge the Dept. of Biotechnology, Govt. of India for partial support through Grant No. BT/PR13244/GBD/27/245/2009. AG acknowledges the Science and Engineering Research Board (SERB), Govt. of India, for partial support through Grant No. EMR/2016/003730.

## Additional information

### Funding

| Funder | Grant reference number | Author |
| --- | --- | --- |
| Department of Biotechnology, Ministry of Science and Technology | BT/PR13244/GBD/27/245/2009 | Pramod Pullarkat Aurnab Ghose |
| Science and Engineering Research Board | EMR/2016/003730 | Aurnab Ghose |

The funders had no role in study design, data collection and interpretation, or the decision to submit the work for publication.

## Author contributions
Sushil Dubey, Conceptualization, Formal analysis, Investigation, Methodology, Writing - original draft, Writing - review and editing; Nishita Bhembre, Formal analysis, Investigation, Methodology; Shivani Bodas, Investigation, Methodology; Sukh Veer, Investigation; Aurnab Ghose, Conceptualization, Supervision, Funding acquisition, Methodology, Writing - review and editing; Andrew Callan-Jones, Conceptualization, Software, Formal analysis, Funding acquisition, Methodology, Writing - original draft, Writing - review and editing; Pramod Pullarkat, Conceptualization, Supervision, Funding acquisition, Investigation, Writing - original draft, Project administration, Writing - review and editing

## Author ORCIDs
Sushil Dubey (iD) https://orcid.org/0000-0002-8964-9062
Aurnab Ghose (iD) http://orcid.org/0000-0002-2053-3918
Pramod Pullarkat (iD) https://orcid.org/0000-0003-2716-7575

## Decision letter and Author response
Decision letter https://doi.org/10.7554/eLife.51772.sa1
Author response https://doi.org/10.7554/eLife.51772.sa2

# Additional files
## Supplementary files
• Transparent reporting form

## Data availability
All data generated or analysed during this study are included in the manuscript and supporting files.

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

## Appendix 1

### Derivation of *Equation 6*

We first recall from the main text that the total axon tension in the model is given by

$$\mathcal{T} = \sum_{i=1}^{M} n_i T_i. \tag{9}$$

where

$$T_i = T_i(\Lambda); \qquad \Lambda \equiv \lambda \ell'_{i0}/\ell_i \tag{10}$$

is the tension acting on an element of type $i$, $\ell'_{i0}$ is the initial end-to-end distance of the element $i$, $\lambda = 1 + \gamma$ is the axon deformation , $\gamma$ and the strain. The rest lengths, $\ell_i(t)$, evolve according to

$$\frac{d\ell_i}{dt} = -v_{if}(\ell_i - \ell_{if}) + v_{iu}(\ell_{iu} - \ell_i), \tag{11}$$

where the folding and unfolding rates are

$$v_{iu} = v_{iu}(0)\, e^{\beta T_i \Delta x_i^{fu}} \tag{12}$$

$$v_{if} = v_{if}(0)\, e^{-\beta T_i \Delta x_i^{uf}}. \tag{13}$$

The quantities $\beta$, $\Delta x_i^{fu}$, and $\Delta x_i^{uf}$ are constants and are defined in the main text.

We suppose that the model axon is initially at steady state, with rest lengths $\{\ell_i^{ss}\}$ obtained by solving *Equation 11* with $d\ell_i/dt = 0$. From this steady-state, a sudden, small change in strain, $\delta\gamma$, is applied at some time $t = t_{step}$. The lengths $\{\ell_i\}$ will change in time and relax to a new steady state. The total tension $\mathcal{T}$ will evolve, too. Linearizing *Equation 11* for small changes $\delta\ell_i$ yields

$$\frac{d\delta\ell_i}{dt} = -(v_{if}^{ss} + v_{iu}^{ss})\delta\ell_i - \delta v_{if}(\ell_i^{ss} - \ell_{if}) + \delta v_{iu}(\ell_{iu} - \ell_i^{ss}). \tag{14}$$

Here, $v_{if}^{ss}$ and $v_{iu}^{ss}$ are the folding and unfolding rates at the steady-state prior to $t_{step}$. Writing $\delta v_{if} = \delta\ell_i \frac{dv_{if}}{d\ell_i}\big|_{ss}$ and $\delta v_{iu} = \delta\ell_i \frac{dv_{iu}}{d\ell_i}\big|_{ss}$, where the derivatives are evaluated at the steady-state prior to $t_{step}$, we find that *Equation 14* can be re-written in the form

$$\frac{d\delta\ell_i}{dt} = -\frac{1}{\tau_i}\delta\ell_i, \tag{15}$$

where the relaxation time $\tau_i$ is

$$\tau_i = \left[ (v_{if}^{ss} + v_{iu}^{ss}) + (\ell_i^{ss} - \ell_{if})\frac{dv_{if}}{d\ell_i}\bigg|_{ss} + (\ell_i^{ss} - \ell_{iu})\frac{dv_{iu}}{d\ell_i}\bigg|_{ss} \right]^{-1}. \tag{16}$$

Thus, after the step, each $\delta\ell_i$ will relax exponentially with characteristic time $\tau_i$. Since each tension $T_i$ is a function of $\ell_i$, the linearized change in tension $T_i$ will also relax exponentially, with the same characteristic time. Thus, the total tension, *Equation 9*, will evolve according to

$$\mathcal{T}(t) = \sum_{i=1}^{M} n_i T_i(t = t_{step})\, e^{-(t - t_{step})/\tau_i}, \tag{17}$$

which is *Equation 6* of the main text.

## Derivation of *Equation 7*

We consider now only one type of elastic element and thus drop the index $i$. From *Equations 12 and 13*,

$$\frac{d\nu_\mathrm{u}}{d\ell} = -\nu_\mathrm{u}\frac{\Lambda K(\Lambda)}{\ell}\beta\Delta x^\mathrm{fu} \tag{18}$$

$$\frac{d\nu_\mathrm{f}}{d\ell} = \nu_\mathrm{f}\frac{\Lambda K(\Lambda)}{\ell}\beta\Delta x^\mathrm{uf}. \tag{19}$$

Here, $K = dT/d\Lambda$. *Equation 16* can then be re-written

$$\tau = \left\{(\nu_\mathrm{f}+\nu_\mathrm{u}) + \beta\frac{\Lambda K(\Lambda)}{\ell}\left[(\ell-\ell_\mathrm{f})\Delta x^\mathrm{uf}\nu_\mathrm{f} + (\ell_\mathrm{u}-\ell)\Delta x^\mathrm{fu}\nu_\mathrm{u}\right]\right\}^{-1}, \tag{20}$$

where it is understood all quantities are evaluated at the steady-state prior to $t_\mathrm{step}$, and we have dropped the superscript ss. Finally, noting that the steady state rest length is given by $\ell = (\ell_\mathrm{u}\nu_\mathrm{u} + \ell_\mathrm{f}\nu_\mathrm{f})/(\nu_\mathrm{u}+\nu_\mathrm{f})$, the above equation, after some algebra, can be simplified to

$$\tau = \left[(\nu_\mathrm{u}+\nu_\mathrm{f}) + \frac{\nu_\mathrm{u}\nu_\mathrm{f}}{(\nu_\mathrm{u}+\nu_\mathrm{f})}\frac{(\ell_\mathrm{u}-\ell_\mathrm{f})}{\ell}\beta(\Delta x^\mathrm{fu}+\Delta x^\mathrm{uf})\Lambda K(\Lambda)\right]^{-1}. \tag{21}$$

This is the expression for the relaxation time given in *Equation 7* of the main text.

## Choice of numerical values used to generate *Figure 5B* and *Figure 5C*

We used a choice a parameters appropriate for spectrin to generate *Figure 5B* and *Figure 5C*. For the persistence length, we used $\ell_p = 20$ nm, as measured for erythrocytes (*Stokke et al., 1986*). For the initial end-to-end extension, we note that the distance between adjacent actin rings in the axon is about 180–190 nm (*Xu et al., 2013*). Accounting for the thickness of a ring, we assumed a value of $\ell_0' = 170$ nm. Next, we used $\ell_\mathrm{f} = 200$ nm as the fully folded rest length of a spectrin tetramer, based on each folded repeat being 5 nm long and there being about 40 repeats along the length of the tetramer. The amount of contour length added upon repeat unfolding is broadly distributed (*Lenne et al., 2000*), but we may assume a value of 25 nm. Doing so, we have $\ell_\mathrm{u} = 1200$ nm.

The transition from the folded to the unfolded state involves crossing an energy barrier, as illustrated in *Figure 5A*, representing the intramolecular bonds that need to be broken. The so-called activation length $\Delta x^\mathrm{fu}$, which is the change in reaction coordinate (aligned with the tension) between the barrier position and the folded minimum, is assumed to be 2.5 nm. This value is consistent with an earlier estimate (*Zhu and Asaro, 2008*). The value of the activation length for the reverse reaction, $\Delta x^\mathrm{uf}$, is expected to be considerably larger, since a significant conformation change of the protein is needed to re-fold. We estimate this to be $\Delta x^\mathrm{uf} \sim \frac{\ell_\mathrm{u}}{\ell_\mathrm{f}}\Delta x^\mathrm{fu} = 17.5$ nm.

Finally, for the unfolding and re-folding rates in the absence of tension, we take $\nu_\mathrm{u}(0) = 10^{-5}$ s$^{-1}$ and $\nu_\mathrm{f}(0) = 100$ s$^{-1}$. These values are similar to those measured in Ref (*Scott et al., 2004*). Given the choice of the other parameters described above, this choice of zero-force rates yielded tension relaxation times comparable to those measured for our axon stretching experiments.

**Appendix 1—table 1.** STED Super Resolution Microscopy analysis of different DIV axons—STED image analysis for quantification of rings: Here we quantify the occurrence of periodic rings (ladders) seen at various DIVs within an imaging field of view. It is evident from the table that as axon mature the rings develop all over the axons.

| DIV | No. of | Regular ladder | | Ladder in patches | | No visible ladder | |
|---|---|---|---|---|---|---|---|
| | Axons | Number | % | Number | % | Number | % |
| 1 | 14 | 3 | 21.4 | 3 | 21.4 | 8 | 57.1 |
| 2 | 13 | 4 | 30.8 | 6 | 46.2 | 3 | 23.1 |
| 3 | 13 | 8 | 61.5 | 5 | 38.5 | 0 | 0 |
| 4 | 10 | 7 | 70.0 | 3 | 30.0 | 0 | 0 |
| 5 | 13 | 10 | 76.9 | 3 | 23.1 | 0 | 0 |

**Appendix 1—table 2.** Crosslink-detachment vs domain unfolding. A summary of different scenario and their expected mechanical response.

| | Model | Strain-softening of steady state moduli | Solid-like steady state | Peak in relaxation time vs strain |
|---|---|---|---|---|
| i | Randomly crosslinked semiflexible polymer gel (Refs [*Broedersz and MacKintosh, 2014*]) | Initial entropic stiffening and subsequent softening due to force dependent crosslink detachment. Crosslink unbinding causes energy dissipation and stress relaxation. | Solid-like in stiffening regime and fluid-like in softening regime. | No |
| ii | Aligned microtubules with force dependent crosslinks (Refs [*Ahmadzadeh et al., 2015*; *de Rooij and Kuhl, 2018*]) | Can exhibit direct softening due to crosslink detachment as entropic effect is minimal. | Fluid-like at long times unless one invokes another parallel structure with permanent crosslinks. | No |
| iii | Force dependent unfolding and force dependent re-folding of spectrin domains. | Exhibits direct strain softening when spectrin is in fully extended configuration (as suggested by 190 nm tetramer spacing and FRET data) Dissipation and stress relaxation due to unfolding of domains. | Solid-like at long times as unfolded domains can take load unlike detached crosslinks. | Peak in relaxation time vs strain because unfolding rate increases with force while refolding rate decreases with force. |

