## [Decision Letter]

Thank you for submitting your article "The axonal actin-spectrin skeleton acts as a tension buffering shock absorber" for consideration by *eLife*. Your article has been reviewed by three peer reviewers, and the evaluation has been overseen by a Reviewing Editor and Anna Akhmanova as the Senior Editor. The reviewers have opted to remain anonymous.

The reviewers and myself found your work interesting and thought provoking. There is however a number of points, both related to the experiments and the theoretical model, that need to be explained before a final decision can be reached regarding its possible publication in *eLife*. Please aim to submit the revised version within two months.

Summary:

This paper presents a new study of the elastic response of axons to stretch, which is applied orthogonally to the axon axis. It is found that the axon does not lose its elastic property, but undergoes softening. Drug treatments indicate that the axon stiffness is most strongly affected by the actin component, rather than the microtubules. These observations are compared to a simple model based on elasticity that is composed of springs, which can soften due to unfolding events and show that this explains the main features of the observations. The softening is argued to occur through the unfolding of spectrin tetramers interconnecting periodically-spaced actin filaments.

Essential revisions:

Experiments:

– The use of cytoskeletal drugs makes of course a lot of sense, but it is surprising that the possibility of an active contraction as tension generation process was not tested by using blebbistatin or other contractility inhibitors. These experiments are important to show that myosin-driven contraction is not the cause of the measured tension.

– The plot of Figure 4C (control) seems to be quite different than the inset of Figure 2D. Should not these be essentially the same? Also, why is the spread of the "Ctl" values in Figure 3D so different for the Nocadozole and Latrunculin experiments?

– The authors note "This modulus is expected to be different from that measured using AFM or magnetic tweezers where the imposed force or strain is radial (Ouyang, Nauman and Shi, 2013; Grevesse et al., 2015)". Please explain why? Regardless, it would be helpful, if the measured modulus could be compared to the reported values.

– Can the authors exclude that there is friction between their cantilever and the substrate? For example, because they scratch the surface. To what extent could/would such a cantilever-substrate interaction lead to similar relaxation effects as observed.

Model:

– Is there an argument to support the assumption that the persistence length is independent on the spectrin unfolding?

– Equation 5 states that the tension relaxation time is dominated by the fastest timescale between folding and unfolding. It is a simplified version of Equation S1, which includes another term related to the change of tension upon (un)folding, which is said to be negligible. It seems rather strange that tension relaxation is dominated by the folding rate for small \Δ R. Looking at the equations, it might be because the system is solved close to the equilibrium condition, while in the real experiment, the stretching describes a non-equilibrium process. Furthermore, the relaxation curve is fitted by a double exponential, and Equation 5 is said to explain while the longest relaxation time shows a maximum for a given strain, but experimentally, a maximum can be seen in the short timescale as well. Why is the relaxation curve not fitted with the full linearised solution of Equation 91-4), (the one leading to Equation S1). Would two characteristic timescales also emerge from such a fit? If not, what is the origin of the short timescale and why does it also display a maximum at finite strain? Why is a quantitative comparison of Figure 2B and 5B not presented?

– Although the theoretical model qualitatively reproduces the experimental observations, there are question regarding the order of magnitudes. While the model seems to require very large strains (Figure 5C), the strains in the experiment are very small. Is this a hint that the actual unfolding of the spectrin in vitro is not as large as known from the single molecule experiments, where >15% strain per opening of spectrin repeats is reported. This difference between the strain in Figure 5C and the strain in the experiments is currently a bit hidden but should be discussed more openly.

Discussion:

– The authors have identified a strain-softening behavior for axons, whereas Peter and Mofrad, 2012, observed a strain-stiffening behavior for axons interconnected with MAP tau proteins (in the absence of the surrounding actin-spectrin structure). A discussion on how the strain-stiffening behavior of axon-tau bundle, as observed previously, fits in the proposed model in the current manuscript would enrich the discussion.

– Notably, Peter and Mofrad, 2012, studied the axonal response in much finer temporal resolution (microsecond timescale) than the timescales presented in the present manuscript. Therefore, authors should discuss how behavior of axons might or might not differ at different timescales.

– How microtubules are connected to the actin-spectrin structure is not fully understood yet. Authors should discuss this in regard to their results, and more specifically explain their assumption in this regard in their model. Is it assumed that the actin-spectrin structure is fully connected to the microtubule bundle?

– The authors should also discuss how the mechanical perturbation imposed in the in-vitro setup reflects potential physiological perturbation, and to what extent are the conclusion of the present paper relevant in-vivo.

– It seems rather likely that the mechanism that described, of softening through unfolding, is not limited to the cortical spectrin network, but it’s actually a property shared by the bulk of the axon cytoplasm. The authors ignore the role of neurofilaments, which fill the bulk of the axon. Such molecules are also composed of many folded domains, and maintain the bulk properties of the axon, see for example:

Beck et al., 2010 and Kornreich, Micha, et al. "Neurofilaments function as shock absorbers: compression response arising from disordered proteins." Physical review letters 117.14 (2016): 148101.‏ The contribution of the neurofilaments and how they may enter the model should be discussed.

– Disrupting the cortical actin-spectrin network was shown to disrupt the MTs, so the cortical and bulk properties seem closely linked in the axon (as discussed in the point above). They therefore cannot conclude that the cortex is responsible for the elastic response on its own. See: Qu et al., 2017.‏

– What may be the contribution of ATP-driven processes in the experiments? It seems likely that motors binding/unbinding and pulling should be relevant at the time scales explored. These processes should be discussed. In particular, it is claimed that the observation of a steady-state tension precludes unbinding processes, which would lead to a long-time viscous response. But this might not be correct is unbinding is coupled of active stress generation. This possibility needs to be discussed.

– The following reference seems to be highly relevant and should be cited:

Zhang et al., 2017.‏

– From the model, one would expect a strong increase in stiffness when all spectrin repeats are stretched out. This was never observed experimentally. Could one not simply go on with the experiments until this moment arises. The length at which this starts should also relate to the number of unfolded spectrin repeats.

– Results are often shown in terms of a Young modulus. The derivation of this modulus from the data involves the axon radius, which is assumed not to change. Since many biological systems have a Poisson ratio close to 0.5 this assumption that the cross section does not change when stretching the axon seems hence not trivial at all.

Either it needs to be shown that the radius remains indeed constant, or, if not constant, include the correct diameter. If unknown, check at least to what extent the measured difference in E could be due to changes in diameter that are not accounted for. An alternative (more practical may be) approach could be to defined an effective neurite spring stiffness independent of the diameter.

---

## [Author Response]

Essential revisions:

We would like to begin by thanking the reviewers, the Reviewing Editor and the Senior Editor for the consolidated report which has helped us to substantially improve the manuscript. We now present experiments that rule out involvement of active actomyosin contractility and have generalised the presentation of the model to include possible contributions from other elements that may also undergo unfolding-refolding kinetics. The latter change also allows us to account for the multiple timescales observed in the experiments. Although the equations have been modified, the main mechanism and conclusion remains the same. We also present an enhanced Discussion based on the several suggestions made by the reviewers. Our main conclusions remain the same and are as follows.

• Axons exhibit reversible strain-softening response and tension buffering responses which may allow them to undergo fast reversible stretch without damage.

• The reversible softening or tension buffering response, solid-like steady state, and the peak in the stress relaxation vs. strain plots, can all be accounted for if we invoke force-induced unfolding-refolding kinetics of protein domains.

• Perturbation experiments allow us to compare the relative importance of microtubules and F-actin. These results clearly bring out the mechanical significance of the recently discovered actin-spectrin periodic skeleton in axonal response to stretch.

Thus, we have used a powerful new technique to reveal unique properties of axonal mechanical response to stretch and have tried to connect them to axonal ultrastructure. These findings are likely to be significant for our understanding of axonal resilience to stretch. We hope that the reviewers and the editors find the revised version acceptable for publication. Below, we are providing a point-by-point response to all the queries raised by the reviewers.

Experiments:– The use of cytoskeletal drugs makes of course a lot of sense, but it is surprising that the possibility of an active contraction as tension generation process was not tested by using blebbistatin or other contractility inhibitors. These experiments are important to show that myosin-driven contraction is not the cause of the measured tension.

We have now performed experiments after exposing axon to the myosin-II inhibitor Blebbistatin and we now present this data (we had checked this earlier, but now we present full data). There is no significant change in the axonal response when actomyosin contractility is inhibited. This is because, in order to explore the passive mechanical response and to suppress active responses and to arrest growth-cone migration, we had chosen to do all measurements at room temperature (as active processes have an Arrhenius dependence on temperature a reduction by about 10 degrees reduces these contributions significantly). This simplifies the interpretation of our data. The blebbistatin data which we now include in the revised version (Figure 3—figure supplement 1, Figure 3—figure supplement 2).

– The plot of Figure 4C (control) seems to be quite different than the inset of Figure 2D. Should not these be essentially the same? Also, why is the spread of the "Ctl" values in Figure 3D so different for the Nocadozole and Latrunculin experiments ?

The plots for control axons in 4C is an average over many axons whereas the ones presented in the inset to 2D are tension vs strain data for individual axons. In 4C we intent to highlight the large difference in steady state tension for normal cells and β-II spectrin knockdown cells. We now present the Figure 4C data for individual axons in Figure 4—figure supplement 5. The individual curves are similar to the ones presented in inset of 2D.

Regarding Figure 3D, the steady state tension T_ss_ after a given strain varies from axon to axon due to variations in structure, diameter or rest tension (unlike Young’s modulus, where axon diameter and rest tension are taken into account, limiting the spread in the data). It is for this reason that we compare T_ss_ data for the same axon before and for two time points after drug treatment. To minimise such axonal variations from affecting the analysis we now normalise the tension values for each axon with the initial tension for Figures 3E,F,G, and retain the raw data in Figure 3D. Both sets of data show that the reduction in steady state tension for each axon is more drastic after Lat-A treatment as compared to Noco. The intrinsic variations from axon to axon is reflected in the difference in distribution of tension values in Figure 3D. The difference in distribution of tension values between the two controls is because of limited number of axons included. This is because drug treated axons become very fragile and detach easily from the substrate limiting the number of successful trials.

– The authors note "This modulus is expected to be different from that measured using AFM or magnetic tweezers where the imposed force or strain is radial (Ouyang, Nauman and Shi, 2013; Grevesse et al., 2015)". Please explain why? Regardless, it would be helpful, if the measured modulus could be compared to the reported values.

The axonal cytoskeleton is highly anisotropic. Microtubules (MT) and neurofilaments (NF) are aligned along the axonal shaft and the one-dimensional actin-spectrin periodicity is also along the axon. For these reasons, like for any anisotropic material, the elastic response to a deformation along the shaft (axon length strain) is expected to be different from a radial deformation. In mathematical terms, the modulus has to be represented by a tensor and not a scalar. The viscoelastic response can also depend on the type of deformation. Consider, for example, a crosslinked MT bundle. Such a bundle will not be able to sustain an extensile stress (radial or longitudinal) for long times as crosslinks detach and release stress with time. However, the bundle can sustain a radial compressive stress (as applied by an AFM tip) over long time. Thus the bundle is fluid-like to extension at long timescales but solid-like to compression even at long times.

Ouyang et al., 2013, used chick DRGs and a rounded AFM tip of 25 micron to compress axons and use Hertz contact mechanics to estimate Young’s modulus. They report that for radial compressive strains, MTs make the maximum contribution to the total modulus, followed by NFs and F-actin (they seem to have labeled their curves wrongly in Figure 5 as it does not match the description and numbers in the main text). They estimate a modulus of about 10 kPa for control axons, but as mentioned by these authors, this method is not well suited for determining absolute modulus (due to large mismatch between axon dia. (~1 um) and bead (25 um), and geometry) but good enough to make comparative studies before and after drug treatments.

Grevesse et al., 2015, have probed the mechanical response of axons of rat cortical neurons using magnetic bead rheometry. Here a fibronectin coated bead is attached to the side of an axon and pulled radially at constant force using an electromagnet. This then is a system where the three components (F-actin, NF and MT) are in series. Using creep compliance measurements, they measure an elastic modulus of 7 kPa for control cells. They conclude that NFs contribute mainly to the viscous part of the response whereas microtubules to the elastic part at these timescales (~minute). They point out that the small persistence length (~150 nm) and the weak attractive bonds between neurofilaments make them predominantly viscous compared to axonal microtubules.

In contrast, our experiments are performed by stretching axons along their lengths where the different cytoskeletal components (actin-spectrin lattice, NF and MT) occur in parallel configuration. In this configuration we are most sensitive to the stiffest component and deletion of this component will cause the most drastic reduction in tension at a given strain or modulus (longitudinal modulus).

We have now clarified this in the revised version and have included the modulus measured by AFM for comparison.

– Can the authors exclude that there is friction between their cantilever and the substrate? For example, because they scratch the surface. To what extent could/would such a cantilever-substrate interaction lead to similar relaxation effects as observed.

Several precautions were taken to eliminate such artefacts. These include tests done on the setup using mock samples, checks before and during each experiment, and post analysis. These are detailed below and now clarified in the Materials and methods.

i) The alignment of the piezo plus associated mounts is adjusted such that the cantilever tip moves parallel to the coverslip. Any possible tilt in the coverslip which can occur from experiment to experiment can change this condition but such tilt is easily noticed by observing changes in focus of tiny debris sticking to the surface of the coverslip, which are always present in a cell culture.

ii) After bringing the fiber tip close to the axon of interest, the cantilever tip is brought down using the z-axis of the motorized stage until it just touches the glass coverslip (revealed by observing the dying out of thermal fluctuations, co-focussing of cantilever tip and junk particles on the surface, change in light spot due to Euler buckling of the cantilever, and/or, when in doubt, by moving the piezo away from the axon and looking at the tip response). The cantilever is then pulled up from the surface and tested for free tip movement. Most axons that are free from the surface and attached only at their extremities (axons that exhibit thermal fluctuations all along their lengths) remain in contact with the cantilever even when the cantilever is moved up several microns from the surface. The objective is then brought up to focus on the tip of the cantilever, which is now away from the glass surface. The entire experiment is then video recorded and post-inspected (nature of movement of the tip and focus of the cantilever image). In addition, we also check for possible slippage of the axonal end points.

iii) Cantilever intermittently rubbing against the glass surface during an experiment will produce tell-tale stick-slip like responses (either no relaxation after a step in case of high friction or discontinuous relaxation in case of stick-slip).

Model:– Is there an argument to support the assumption that the persistence length is independent on the spectrin unfolding?

We acknowledge that the persistence length could depend on the state of folding. However, for simplicity we assumed a constant persistence length, which we now mention in the model section. We do not expect that taking this into account would change the qualitative results of our model, in particular softening due to unfolding and a non-monotonic tension relaxation time.

– Equation 5 states that the tension relaxation time is dominated by the fastest timescale between folding and unfolding. It is a simplified version of Equation S1, which includes another term related to the change of tension upon (un)folding, which is said to be negligible. It seems rather strange that tension relaxation is dominated by the folding rate for small \Δ R. Looking at the equations, it might be because the system is solved close to the equilibrium condition, while in the real experiment, the stretching describes a non-equilibrium process.

Indeed, it might seem surprising that tension relaxation is dominated by the folding rate (and not the unfolding rate) of a protein crosslinker at low tension, given that most domains are folded. However, we can explain this by referring to Equation 3 of the revised manuscript. This is a rate equation for the protein’s contour length, l(t): dl/dt=-nu_f_*(l-l_f_)+nu_u_*(l_u-l_). Here, nu_f_ and nu_u_ are the folding and unfolding rates, respectively; l_f_ is the contour length in the fully folded state; and l_u_ is the contour length in the fully unfolded state.

The rates are functions of l(t) via Boltzmann factors. Also, l-l_f_ is proportional to the number of unfolded domains on the proteins, and l_u-l_ is proportional to the number of folded domains. Now, when a sudden perturbation from steady-state (dl/dt=0) is applied, l(t) will relax to some new steady-state. If the perturbation is small, we can linearize the rate equation. The relaxation will then be governed by d(δ l)/dt=(nu_f_+nu_u_)*δ l+…, where the omitted terms are not important for this argument. Here, nu_f_ and nu_u_ are constants, evaluated at the steady-state prior to the perturbation. Thus, the relaxation time will involve both nu_u_ and nu_f_, independently of the number of unfolded domains in the steady state prior to the perturbation. Furthermore, for small tension, most domains are indeed folded, and hence nu_f_>>nu_u_, and thus nu_f_ dominates the tension relaxation.

We feel that the linearization around steady-state provides a good description of the experimental tension relaxation. In particular, the tension relaxation data were fitted well with the sum of two exponential functions; exponential relaxation is only expected in the linearized regime.

Furthermore, the relaxation curve is fitted by a double exponential, and Equation 5 is said to explain while the longest relaxation time shows a maximum for a given strain, but experimentally, a maximum can be seen in the short timescale as well. Why is the relaxation curve not fitted with the full linearised solution of Equation 91-4), (the one leading to Equation S1). Would two characteristic timescales also emerge from such a fit? If not, what is the origin of the short timescale and why does it also display a maximum at finite strain? Why is a quantitative comparison of Figure 2B and 5B not presented?

In the earlier version of the model, which considered only one type of protein crosslinker (spectrin) that could unfold and refold, there was a single relaxation time. The approximation of neglecting the second term on the right of Equation S1 in earlier version, leading to Equation 5 in earlier version, does not change this fact. Thus, two or more relaxation times would not emerge from fitting the tension data to the full linearized solution.

In the revised model, in which multiple protein crosslinkers act in parallel, there is a distinct tension relaxation time coming from each type of crosslinker. Moreover, each relaxation time has the same generic dependence on strain, and we expect non-monotonic behavior to occur quite generally. Our fits of the tension relaxation data using the sum of two exponentials suggest that two types of crosslinkers dominate. It is difficult to know which crosslinkers the slow (tau_1_) and fast (tau_2_) relaxation times correspond to. We might speculate that since tau proteins have little secondary structure and can thus unfold fairly easily, that these crosslinkers correspond to the fast relaxation time.

Finally, we chose not to fit the tension relaxation data (2B) with the model (5B) because the model contains a certain number of parameters that are not well known in the context of axons. The purpose of the model is not to accurately fit the tension relaxation data, thereby obtaining estimates of the various fit parameters, but to provide physical insight into the problem. We have indicated in a few instances in the manuscript the qualitative character of the predictions of the model.

*– Although the theoretical model qualitatively reproduces the experimental observations, there are question regarding the order of magnitudes. While the model seems to require very large strains (Figure 5C), the strains in the experiment are very small. Is this a hint that the actual unfolding of the spectrin* in vitro *is not as large as known from the single molecule experiments, where >15% strain per opening of spectrin repeats is reported. This difference between the strain in Figure 5C and the strain in the experiments is currently a bit hidden but should be discussed more openly.*

We agree with the reviewer that the range of strain explored in the earlier version of the model (Figure 5) was not very representative of what is accessible experimentally. We have now modified Figure 5 so that the range of axon strain does not exceed 10% . Significant unfolding occurs, even for small strains, because the protein crosslinkers in the revised model are under tension before the axon is strained.

Discussion:– The authors have identified a strain-softening behavior for axons, whereas Peter and Mofrad, 2012, observed a strain-stiffening behavior for axons interconnected with MAP tau proteins (in the absence of the surrounding actin-spectrin structure). A discussion on how the strain-stiffening behavior of axon-tau bundle, as observed previously, fits in the proposed model in the current manuscript would enrich the discussion.

Peter et al. have used computer simulation to investigate the mechanical response of microtubule bundles cross-linked by tau. Unlike Rooij and Kuhl (Biophys. J., vol.114, Yr. 2017; DOI: 10.1016/j.bpj.2017.11.010), this model does not consider the possibility of crosslink detachment, or any other mechanism for tension relaxation (such as unfolding). Furthermore, they do not consider the actin-spectrin skeleton.

Therefore, it is not surprising that their model gives a very different response compared to what we see in our experiments. We now elaborate on these differences in the revised version.

– Notably, Peter and Mofrad, 2012, studied the axonal response in much finer temporal resolution (microsecond timescale) than the timescales presented in the present manuscript. Therefore, authors should discuss how behavior of axons might or might not differ at different timescales.

The data presented by Peter et al. are from computer simulations and hence they can access microsecond timescales. Unfortunately, our experimental setup cannot achieve such temporal resolution. Like most viscoelastic material, we expect the axonal response to have time or (frequency) dependence. In the domain unfolding picture, one may expect an increased Young’s modulus at deformation rates faster than the typical unfolding rate. However, the problem is subtle since the unfolding rate depends, of course, on deformation (more precisely, tension). It is an interesting area of study, but is beyond the scope of the current work as data is lacking.

– How microtubules are connected to the actin-spectrin structure is not fully understood yet. Authors should discuss this in regard to their results, and more specifically explain their assumption in this regard in their model. Is it assumed that the actin-spectrin structure is fully connected to the microtubule bundle?

To our knowledge, there is no direct connection reported between the axonal actinspectrin lattice and microtubules. There could be indirect mechanical mechanical coupling, via neurofilaments, for example, which we cannot rule out. We have assumed neurofilaments in axons to be predominantly viscous based on Grevesse et al., 2015, and based on the fact that organelles which are transported on microtubules move easily through this layer (see images and discussion in Safinya et al., Annu.Rev. Condens. Matter Phys. Yr. 2015; doi:10.1146/annurev-conmatphys-031214-014623). In the revised model we consider multiple cytoskeletal structures acting in parallel — such as actin-spectrin lattice and microtubules interconnected by MAPs. This implies some connectivity between them, though the precise nature of these connections is not crucial for the main results of the model.

– The authors should also discuss how the mechanical perturbation imposed in the in-vitro setup reflects potential physiological perturbation, and to what extent are the conclusion of the present paper relevant in-vivo.

The strain range explored in this paper is within what has been reported to occur in vivo during normal limb movements and shear deformations of the brain (values mentioned in the Introduction). Taken together, our experiments and model show that reversible and force dependent protein unfolding events can be a major source of energy dissipation in axons and may aid in protecting axons against sudden rise in tension under normal physiological conditions. These processes increases the threshold for axonal damage. Our identification of spectrin as a major element for this mechanism is also corroborated by studies in *C. elegans* where axons lacking spectrin snap during the wiggling of the worm. We have added a sentence to the Discussion to bring out these potential physiological relevance.

– It seems rather likely that the mechanism that described, of softening through unfolding, is not limited to the cortical spectrin network, but its actually a property shared by the bulk of the axon cytoplasm. The authors ignore the role of neurofilaments, which fill the bulk of the axon. Such molecules are also composed of many folded domains, and maintain the bulk properties of the axon, see for example:Beck et al., 2010 and Kornreich, Micha, et al. "Neurofilaments function as shock absorbers: compression response arising from disordered proteins." Physical review letters 117.14 (2016): 148101.‏ The contribution of the neurofilaments and how they may enter the model should be discussed.

The reviewer rightly points out that spectrin may not be the only cytoskeletal crosslinker in axons that undergoes unfolding and re-folding events. Following the reviewers comments we have generalized the theoretical model to include multiple protein crosslinkers acting in parallel. This leads to multiple tension relaxation times. We then argue that a major contribution to the unfolding-refolding process may come from the actin-spectrin skeleton as this explains drastic fall in modulus whenever this skeleton is perturbed (either by Lat-A or by spectrin morpholino). Our results are further supported by experiments on *C. elegans* lacking spectrin where the axons break easily when the worm wiggles (J. Cell Biol., Vol. 176, No. 3 Yr. 2007;doi: 10.1083/jcb.200611117).

The mechanical contributions of neurofilaments (NFs) to the axon stretch response is far from clear. Magnetic bead rheology experiments by Grevesse et al., 2015, show that NFs are more viscous-like compared to microtubules. The ease with which organelles embedded in neurofilaments are transported in axons too suggest that NFs may be in a fluid-like state (see images and discussion in Safinya et al., Annu.Rev. Condens. Matter Phys., Vol. 6, Yr. 2015; DOI: 10.1146/annurev-conmatphys-031214-014623). For these reasons, we had assumed that neurofilaments do not play a role in supporting axon tension. Now, as we discussed above, in the updated model we have allowed for the possibility of other proteins undergoing unfolding, such as NFs, which may contribute to the axon’s stretch response. We thank the reviewer for the articles he/she has mentioned and have added a paragraph to the Discussion on the possible role played by neurofilaments in axon stretching. We also distinguish compression (as in AFM experiments) from stretching, and the NF responses can be very different in these two cases as elaborated above in response to an earlier query.

– Disrupting the cortical actin-spectrin network was shown to disrupt the MTs, so the cortical and bulk properties seem closely linked in the axon (as discussed in the point above). They therefore cannot conclude that the cortex is responsible for the elastic response on its own. See: Qu et al., 2017.‏

Indeed. We now refer to this paper which reports this connection in fly neurons. However, there are structural differences between vertebrate and fly neurons, for example, fly neurons lack neurofilaments. This potentially allow the spectrin lattice to directly couple to microtubules. In any case, to rule out such coupling affecting our data, we performed combination drug experiments where we first stabilised microtubules with Taxol and then disrupted F-actin (Figure 3G). This way, we ensure that the reduction in steady state tension after disruption of the actin-spectrin lattice is not due to bulk depolymerisation of microtubules.

– What may be the contribution of ATP-driven processes in the experiments ? It seems likely that motors binding/unbinding and pulling should be relevant at the time scales explored. These processes should be discussed. In particular, it is claimed that the observation of a steady-state tension precludes unbinding processes, which would lead to a long-time viscous response. But this might not be correct is unbinding is coupled of active stress generation. This possibility needs to be discussed.

This is a very important point and was not addressed adequately in the earlier version. We and others have shown that axons exhibit acto-myosin contractility (Sampada et al., Bernal et al., Tofangchi et al., all in manuscript). This makes it difficult to separate the active and passive responses as they are not well separated in timescales. In order to suppress active responses from interfering with passive responses, we chose to do all experiments at room temperature (since active processes typically have an Arrhenius response with temperature). To further rule out acto-myosin activity, we have now performed experiments using Blebbistatin and the data shows that the response of treated axons are similar to that of room temperature control (Figure 3—figure supplement 3, Figure 3—figure supplement 4). This data and discussion are now included. We now plan to investigate the mechanical response of active axons and extend the passive model to one that includes activity and this will be part of a future publication.

– The following reference seems to be highly relevant and should be cited:Zhang et al., 2017.‏

Indeed! We have now included this important and interesting computational model for the axonal spectrin skeleton.

– From the model, one would expect a strong increase in stiffness when all spectrin repeats are stretched out. This was never observed experimentally. Could one not simply go on with the experiments until this moment arises. The length at which this starts should also relate to the number of unfolded spectrin repeats.

Yes, the model predicts a stiffening response once all, or a large fraction of the domains are in the unfolded state. We are unable to explore this regime because one of the ends of the axon (usually the cell body which is very weakly anchored) detaches before such high strains could be reached. We are hoping to do this in future studies by developing techniques to hold the cell body in place (perhaps using a micropipette). In addition to providing yet another test for the model, this regime becomes interesting from an axonal injury perspective.

– Results are often shown in terms of a Young modulus. The derivation of this modulus from the data involves the axon radius, which is assumed not to change. Since many biological systems have a Poisson ratio close to 0.5 this assumption that the cross section does not change when stretching the axon seems hence not trivial at all.Either it needs to be shown that the radius remains indeed constant, or, if not constant, include the correct diameter. If unknown, check at least to what extent the measured difference in E could be due to changes in diameter that are not accounted for. An alternative (more practical may be) approach could be to defined an effective neurite spring stiffness independent of the diameter.

We had already performed such an analysis. The change in radius within 15% strain is below the resolution of the microscope. However, we can estimate the change in radius due to stretch. For this, we had earlier assumed that the axonal volume is conserved and then estimated the change in radius arising due to stretch and the error in Young’s modulus from this effect. By doing this we estimated a decrease in radius of about 7% for 15% strain. This small decrease in radius means we slightly *underestimate* the Young's modulus, *E*, and thus cannot account for the softening we observe.

These estimates are added to the revised version under Materials and methods. We prefer using Young’s modulus where possible and this reduces variations from axon to axon due to differences in rest tension and axon initial diameter. Wherever drug treatments are used, we compare tension for a given strain (which is proportional to spring constant) before and after treatment for the same axon and same strain value.